# Affinity maturation generates pathogenic antibodies with dual reactivity to DNase1L3 and dsDNA in systemic lupus erythematosus

Eduardo Gomez-Bañuelos [1,5], Yikai Yu [2,5], Jessica Li[1], Kevin S. Cashman[3], Merlin Paz[1], Maria Isabel Trejo-Zambrano[1], Regina Bugrovsky[3], Youliang Wang[3], Asiya Seema Chida[3], Cheryl A. Sherman-Baust [4], Dylan P. Ferris[1], Daniel W. Goldman [1], Erika Darrah [1], Michelle Petri [1], Iñaki Sanz [3] & Felipe Andrade [1] ✉

Anti-dsDNA antibodies are pathogenically heterogeneous, implying distinct origins and antigenic properties. Unexpectedly, during the clinical and molecular characterization of autoantibodies to the endonuclease DNase1L3 in patients with systemic lupus erythematosus (SLE), we identified a subset of neutralizing anti-DNase1L3 antibodies previously catalogued as anti-dsDNA. Based on their variable heavy-chain ($V_H$) gene usage, these antibodies can be divided in two groups. One group is encoded by the inherently autoreactive $V_H$4-34 gene segment, derives from anti-DNase1L3 germline-encoded precursors, and gains cross-reactivity to dsDNA – and some additionally to cardiolipin – following somatic hypermutation. The second group, originally defined as nephritogenic anti-dsDNA antibodies, is encoded by diverse $V_H$ gene segments. Although affinity maturation results in dual reactivity to DNase1L3 and dsDNA, their binding efficiencies favor DNase1L3 as the primary antigen. Clinical, transcriptional and monoclonal antibody data support that cross-reactive anti-DNase1L3/dsDNA antibodies are more pathogenic than single reactive anti-dsDNA antibodies. These findings point to DNase1L3 as the primary target of a subset of antibodies classified as anti-dsDNA, shedding light on the origin and pathogenic heterogeneity of antibodies reactive to dsDNA in SLE.

Systemic lupus erythematosus (SLE) is a complex multisystem disease characterized by the production of antibodies to a diverse number of autoantigens leading to immune-mediated tissue damage[1]. According to the clonal selection theory of antibody production, the different autoantigen reactivities in SLE serum should be explained by the simultaneous presence of different autoantibodies, each of them with a unique specificity[2]. Yet, growing evidence—including the analysis of patient-derived monoclonal antibodies—has shown that the autoantibody landscape in SLE is populated by pathogenic autoantibodies reacting with multiple antigens[3–15]. Identifying the antigenic cross-reactivity of these autoantibodies has been of interest to investigators trying to elucidate both the potential triggering and the target

[1]Division of Rheumatology, The Johns Hopkins University School of Medicine, Baltimore, MD 21224, USA. [2]Department of Rheumatology, Tongji Hospital, Tongji Medical College, Huazhong University of Science and Technology, Wuhan, Hubei 430074, P. R. China. [3]Department of Medicine, Division of Rheumatology, Lowance Center for Human Immunology, Emory University, Atlanta, GA 30322, USA. [4]Gene Regulation Section, Laboratory of Molecular Biology and Immunology, National Institute on Aging, Baltimore, MD 21224, USA. [5]These authors contributed equally: Eduardo Gomez-Bañuelos, Yikai Yu. ✉e-mail: andrade@jhmi.edu

antigens in SLE. Anti-dsDNA antibodies are of particular interest in this regard. Although the presence of antibodies to dsDNA is a unifying feature among patients with SLE, not all anti-dsDNA antibodies are pathogenic[1,3,16]. These findings have suggested that anti-dsDNA antibodies comprise a heterogeneous pool of autoantibodies with distinct origins, physicochemical and antigenic properties[17,18], and that fine specificities, in addition to dsDNA binding govern their pathogenic effect in SLE[3]. For instance, a subset of anti-dsDNA antibodies that bind the N-methyl-D-aspartate receptor can drive neuronal death and neuropsychiatric lupus[11], and cross-reactivity with intrinsic renal antigens, such as α-actinin, has been proposed as a mechanism by which a subset of anti-dsDNA antibodies can mediate nephritis[15,19]. The basis of the heterogeneity and cross-reactivity of anti-dsDNA antibodies, however, remains unknown.

DNase1L3 is a member of the DNase1 family of DNA endonucleases, which was recently found to be the target of autoantibodies in SLE[20,21]. The enzyme is primarily secreted by myeloid cells (i.e. macrophages and dendritic cells)[22–24], and together with DNase1, is responsible for the DNase activity in circulation[25]. Different to DNase1, however, DNase1L3 is more efficient in the inter-nucleosomal cleavage of nuclear DNA, suggesting that its major role is the digestion of chromatin from apoptotic and necrotic cells and therefore, in regulating the load of immunogenic DNA[26,27]. This notion is supported by the finding that null mutations and hypomorphic variants of DNase1L3 are linked to familial and sporadic SLE, respectively[28,29]. In addition, lupus-prone MRL and NZB/W F1 mice are deficient in DNase1L3, and the sole deficiency of this enzyme leads to lupus-like disease in mice[27,30]. Mechanistically, DNase1L3 decreases the availability of antigenic cell-free DNA by fragmenting DNA, reducing its exposure on apoptotic cell microparticles[27,31]. In the absence of DNase1L3 activity, extracellular self DNA drives TLR-dependent type-I interferon (IFN-I) production and extrafollicular differentiation of antibody-forming cells, driving anti-dsDNA antibodies and SLE[32]. Interestingly, the recent finding that patients with SLE have autoantibodies to DNase1L3 highlights that the DNase1L3 pathway is also pathogenically targeted in sporadic SLE[20,21].

In this work, we combined clinical and blood transcriptional data, together with an extensive analysis of patient-derived monoclonal antibodies, to understand the origin and immunopathology related to anti-DNase1L3 antibodies in SLE. We found that antibodies to DNase1L3 and dsDNA are associated with both clinical and transcriptional features of SLE disease activity. However, it was intriguing that this association was only significant in patients positive for both autoantibodies compared to patients single positive for either antibody. Through the analysis of SLE serum and patient-derived monoclonal antibodies, we found that a subset of anti-DNase1L3 antibodies is also reactive with dsDNA, providing a rational explanation that SLE disease activity is associated with a single autoantibody with dual reactivity to DNase1L3 and dsDNA. Indeed, these autoantibodies are highly mutated, originate from both autoreactive and non-autoreactive precursors, and have the dual ability of neutralizing DNase1L3 activity and bind to dsDNA with high efficiency, likely amplifying their pathogenic potential compared to monospecific autoantibodies. Based on the binding efficiency of mutated and germline reverted monoclonal antibodies to DNase1L3 and dsDNA, the data support that DNase1L3 is the primary target of these antibodies and dsDNA is the cross-reactive antigen. Collectively, our studies underscore DNase1L3 as the protein antigen and functional target of a subset of pathogenic antibodies catalogued as anti-dsDNA in SLE.

## Results
### Anti-DNase1L3 antibodies are associated with clinical and immunological features of active SLE
To define clinical features and SLE-associated immune pathways linked to anti-DNase1L3 antibodies in patients with SLE, we studied a prospective observational cohort for which extensive clinical and serologic variables are available, as well as whole-blood gene expression data[33]. Demographic, clinical and laboratory features of the SLE cohort are summarized in Supplementary Table 1. Antibody reactivity to DNase1L3 was significantly increased in SLE compared to healthy controls (P < 0.001) (Fig. 1a and Supplementary Fig. 1). Using a cutoff of two standard deviations above the mean anti-DNase1L3 antibody level in healthy sera, 30% (48/158) of SLE patients *versus* 1.6% (1/62) of healthy controls were positive for anti-DNase1L3 antibodies (P < 0.0001). Antibodies to DNase1L3 were significantly associated with anemia, livedo, proteinuria, low complement levels, use of cytotoxic treatment, and a broad range of autoantibodies, including anti-dsDNA, anti-cardiolipin, lupus anticoagulant, anti-β2-glycoprotein I (B2GPI), anti-Ro52 antibodies (Fig. 1b and Supplementary Table 1).

At the time of the visit, anti-DNase1L3 antibodies were significantly associated with higher disease activity by SELENA-SLEDAI [median, (IQR), 1.8 (0–12) vs. 3.4 (0–12), P = 0.002] (Fig. 1c), which was determined by the immunological domain (Fig. 1d). Specifically, elevated anti-dsDNA [12.7% (14/110) vs. 52.1% (25/48), P < 0.0001] and low complement [6.4% (7/110) vs. 31.2% (15/48), P < 0.0001] were more common with anti-DNase1L3 antibodies (Fig. 1e). Further, at time of visit, multivariate analyses showed that anti-DNase1L3 positive patients were more likely treated with prednisone [OR (95% CI) 2.6 (1.23–5.42)] and cytotoxic drugs [OR (95% CI) 4.6 (2.16–9.89)] independently of disease activity (Supplementary Table 2).

### Antibodies to DNase1L3 are associated with transcriptional fingerprints linked to disease activity in SLE
Patients with SLE display unique blood transcriptional profiles, including hallmark signatures linked to immune dysregulation[34]. To define whether antibodies to DNase1L3 are associated with distinct transcriptional fingerprints in SLE, we used gene expression data from blood collected in parallel with the samples used to measure anti-DNase1L3 antibodies. We identified 584 differentially expressed transcripts (DETs) between anti-DNase1L3 negative and positive SLE patients (Fig. 2a). Using unsupervised hierarchical clustering of the 584 DETs, SLE patients clustered in three major groups defined by the expression of IFN-stimulated genes (ISGs), inflammation, neutrophil activation genes, and genes related to mRNA processing and translation (Fig. 2b, c). Strikingly, 79% of patients positive for anti-DNase1L3 antibodies clustered with overexpression of ISGs and neutrophil activation genes (Cluster Three, P < 0.0001) (Fig. 2d). Moreover, comparison of the microenvironment cell populations (MCP)-counter[35] deconvolution score for T-cells, monocytes and neutrophils showed a significant increase in neutrophil counts in anti-DNase1L3 positive, compared with negative patients [median MCP-score (IQR), 1319 (1187–1472) vs. 1229 (1104–1330), respectively, P = 0.025] (Fig. 2e), supporting the association of anti-DNase1L3 antibodies with higher disease activity in SLE[36].

To further define whether anti-DNase1L3 antibodies associate with functional gene sets mechanistically linked to SLE[34], we conducted single-sample Gene Set Enrichment Analysis (ssGSEA)[37] of the blood transcription modules described by Chaussabel et al.[38]. Interestingly, modules known to be enriched in SLE[34] were associated with anti-DNase1L3 positivity (Fig. 3a and Supplementary Tables 3 and 4). In particular, the IFN (i.e., M1.2, M3.4, and M5.12), myeloid (i.e., M5.15, and M7.16), inflammation (i.e. M4.2, M4.6, and M5.1), and apoptosis (M6.6 and M6.13) modules were most enriched in the anti-DNase1L3 antibody positive group compared with the anti-DNase1L3 antibody negative group (Fig. 3a and Supplementary Table 5).

Since the IFN modules M1.2, M3.4, and M5.12 have been previously shown to correlate with SLE disease activity[39], we addressed whether the transcriptional signatures of IFN signaling and myeloid activation reflect unique aspects of anti-DNase1L3 and not only an epiphenomenon related to active disease. Since the immunological domain of

SLEDAI (i.e. dsDNA binding, and low complement) was the only component significantly associated with anti-DNase1L3 positivity (Fig. 1d, e), we further compared the module activity according to anti-DNase1L3 and anti-dsDNA status. Multivariate regression analysis showed that anti-DNase1L3 and anti-DNA positivity were independently associated with the IFN modules after adjustment for clinical disease activity, neutrophil count and the use of prednisone or cytotoxic drugs (Supplementary Tables 6 to 8). Nevertheless, only SLE patients positive for both anti-DNase1L3 and anti-dsDNA showed a significant increase of the IFN modules when compared to SLE patients single-positive for anti-DNase1L3 or anti-dsDNA, or negative for both immunoreactivities (Fig. 3b). These findings strongly suggest that the presence of both antibodies has an additive effect on the production of IFN in SLE.

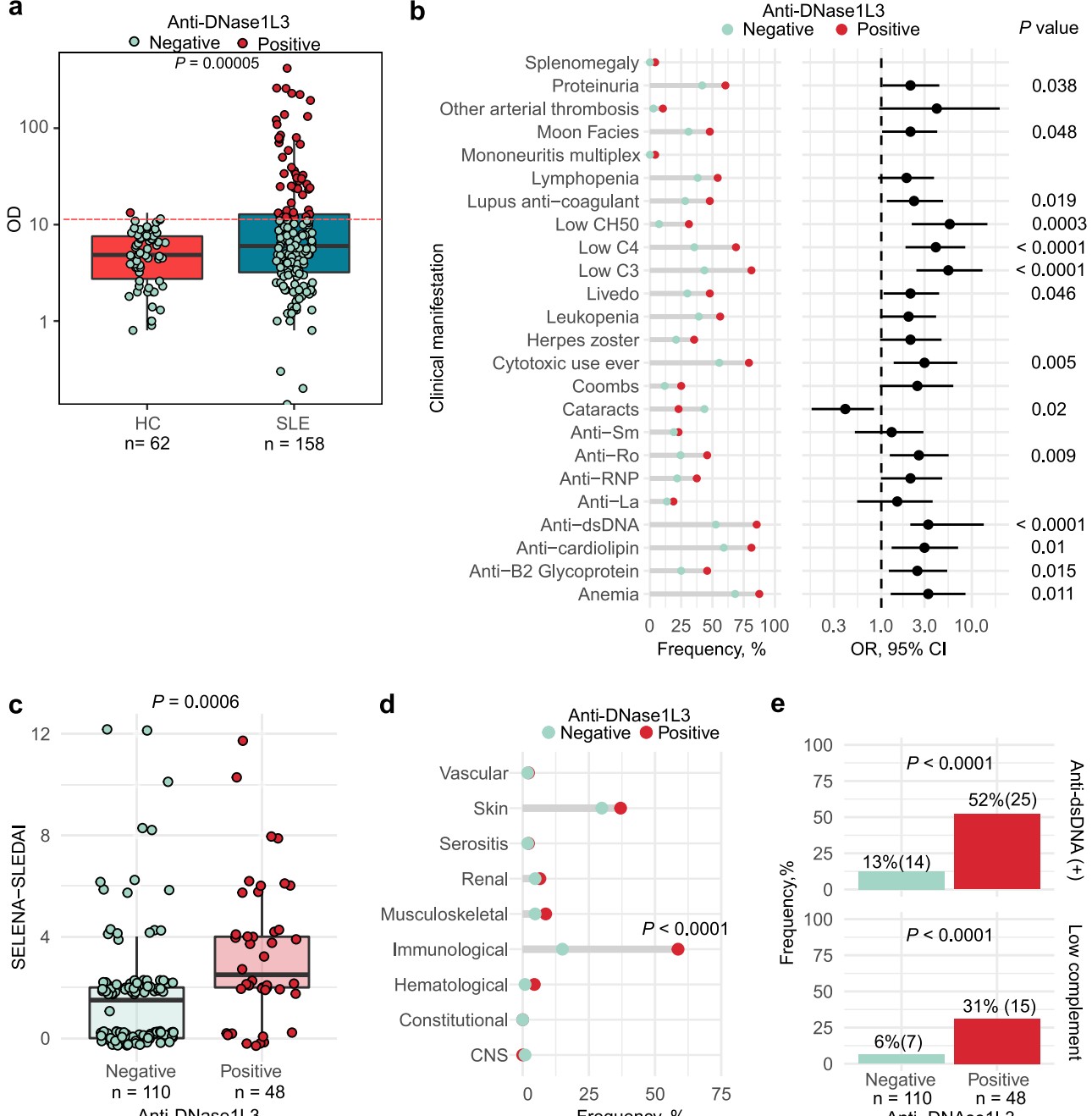

**Fig. 1 | Prevalence, clinical and serologic associations of antibodies to DNase1L3 in a prospective observational cohort of patients with SLE. a** Serum levels and positivity of anti-DNase1L3 antibodies in healthy controls (HC) and patients with SLE. Using a cutoff of two standard deviations above the mean anti-DNase1L3 antibody level in healthy sera, 1.6% (1/62) of healthy controls and 30% (48/158) of SLE patients were positive for anti-DNase1L3 antibodies. **b** Associations between anti-DNase1L3 antibodies and clinical/serologic features in SLE. Bars represent the frequency of clinical/serologic features according to anti-DNase1L3 antibody status (left) and their corresponding OR with 95% CI (right). **c** Safety of Estrogens in Lupus National Assessment study-SLE disease activity index (SELENA-SLEDAI) of SLE subjects, at time of visit, according to anti-DNase1L3 positivity. **d** Associations between individual SELENA-SLEDAI score items 'at time of visit' and anti-DNase1L3 antibodies. SELENA-SLEDAI items are represented as categorical variables. CNS central nervous system. **e** Frequency of SLE subjects positive for anti-dsDNA antibodies (upper) and low complement 'at time of visit' according to anti-DNase1L3 status. Comparisons of continuous variables were done using Student's τ test. Associations between categorical variables were performed with χ² of Fisher's exact tests accordingly.

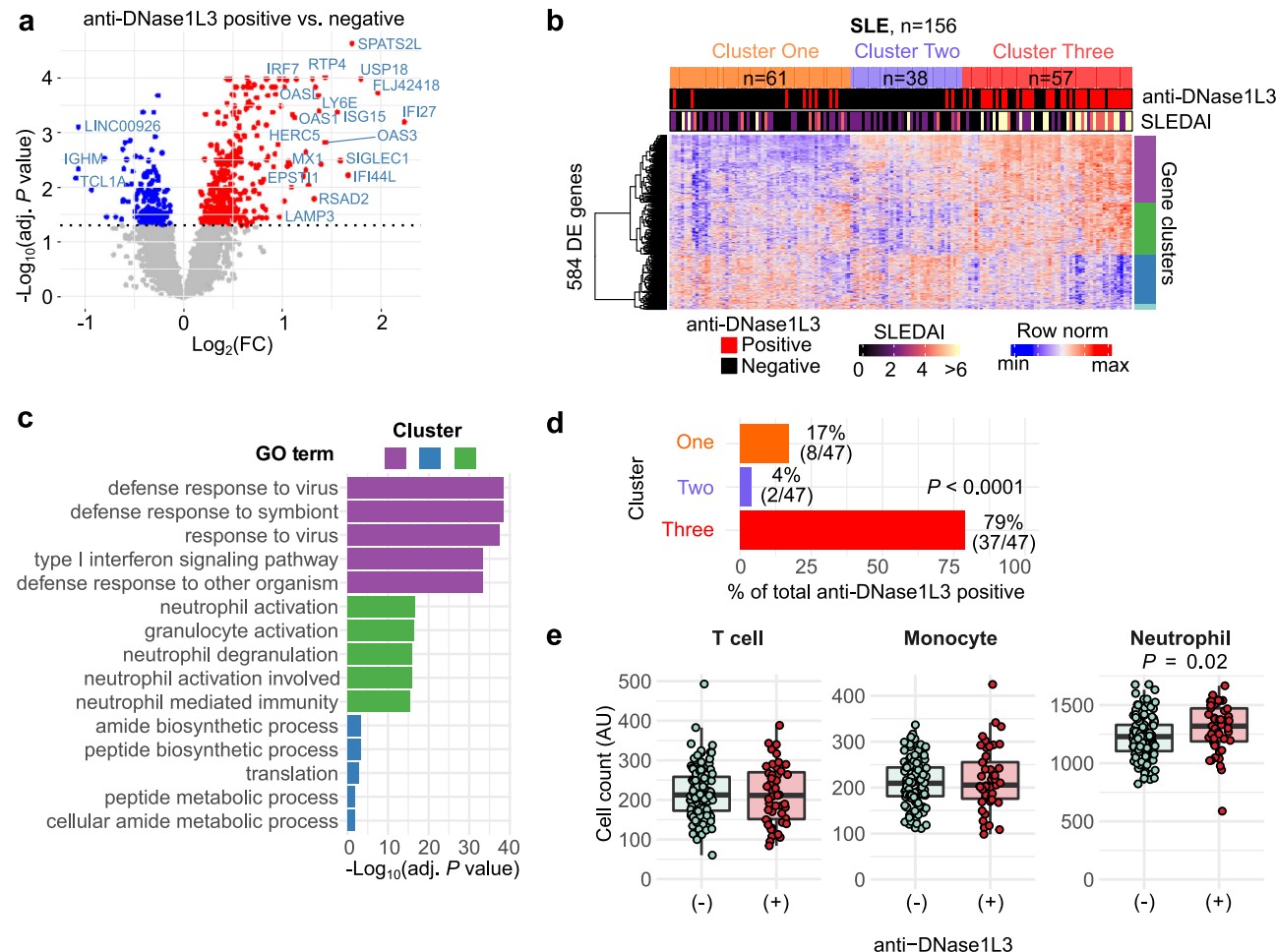

**Fig. 2 | Transcriptional correlates of anti-DNase1L3 antibodies in SLE. a** Volcano plot of 584 differentially expressed transcripts (DETs) between SLE patients positive and negative for anti-DNase1L3 antibodies. Red, 399 upregulated DETs with adjusted $P < 0.01$. Blue, 185 downregulated DETs with adjusted $P < 0.01$. DETs with $P$ value $< 0.01$ and Log2(Fold change) $>1$ are marked (**b**). Hierarchical clustering of 584 DETs from **a**. Each column represents an individual patient and each row an individual gene. Top annotations show cluster membership, anti-DNase1L3 antibodies (positive = red, negative = black), and SLEDAI score in continuous scale. DETs were split in $k = 4$ expression clusters and annotated by functional enrichment analysis using the g: Profiler toolset with the gene oncology molecular function (GO:MF) gene set collection. Red represents upregulated genes and blue downregulated genes. Only 156/158 of SLE patients from Fig. 1a had paired microarray and serum data. SLE patients clustered in three major groups defined by upregulation of genes related to mRNA processing and translation (Cluster One), upregulation of mRNA processing, translation, and some IFN-stimulated genes (ISGs) (Cluster Two), and upregulation of ISGs and neutrophil activation genes (Cluster Three). **c** Top 5 enriched GO:MF terms on gene expression clusters according to $P$ value. **d** Frequency of anti-DNase1L3 antibody positive SLE patients according to cluster membership in **b**. **e** Comparison of the MCP counter deconvolution score for T-cells, monocytes and neutrophils, between anti-DNAse1L3 positive ($n = 47$) vs. anti-DNAse1L3 negative ($n = 109$) SLE. GO gene ontology, FC fold-change.

Similar to the IFN modules, anti-DNase1L3, and anti-dsDNA were independently associated with the myeloid lineage module M7.16 after adjustment for neutrophil count, treatment, and clinical activity (Fig. 3c and Supplementary Table 9). However, the myeloid module M5.15 was only significantly elevated on the double-positive group when compared against double-negative patients (Fig. 3c). Further multivariate analyses concluded that the activity of M5.15 is not independently associated with anti-DNase1L3 or anti-dsDNA antibodies, but with clinical SLEDAI, prednisone use and neutrophil count (Supplementary Table 10). Interestingly, the principal component analysis (PCA) projection of disease activity, blood transcription modules (M1.2, M3.4, M5.12, M.5.15, and M7.16) and neutrophil count, showed that anti-DNase1L3 antibodies identify a subset of SLE patients with higher disease activity, neutrophil count and IFN/myeloid activation (Fig. 3d), which is also characterized by positivity for anti-dsDNA antibodies (Fig. 3f). In the absence of antibodies to DNase1L3, however, the distribution of SLE patients is heterogeneous irrespective of their anti-dsDNA antibody status (Fig. 3e, g). Taken together, these data are consistent with clinical and laboratory features demonstrating that anti-DNase1L3 antibodies are associated with enhanced stimulation of immune pathways activated by cell-free DNA[40,41]. Moreover, the data further support an additive effect between anti-DNase1L3 and anti-dsDNA positivity on clinical and transcriptional markers related to SLE disease activity.

## A subset of antibodies to DNase1L3 arise from autoreactive $V_H4$-34-expressing IgG-switched memory B cells

The autoantibody compartment in SLE is importantly shaped by the expansion of autoreactive B cells using the immunoglobulin variable heavy-chain gene segment $V_H4$-34 (i.e., $V_H4$-34$^+$ B cells)[7,42–45]. To assess whether serum reactivity to DNase1L3 is linked to antibodies containing the $V_H4$-34 encoded heavy chain, antibodies in serum were depleted using the anti-idiotypic antibody 9G4, a monoclonal antibody that specifically targets $V_H4$-34 encoded antibodies[46]. Interestingly, depletion of antibodies bearing the 9G4 idiotype decreased the reactivity to DNase1L3 by 0% to 80% in anti-DNase1L3 positive SLE sera

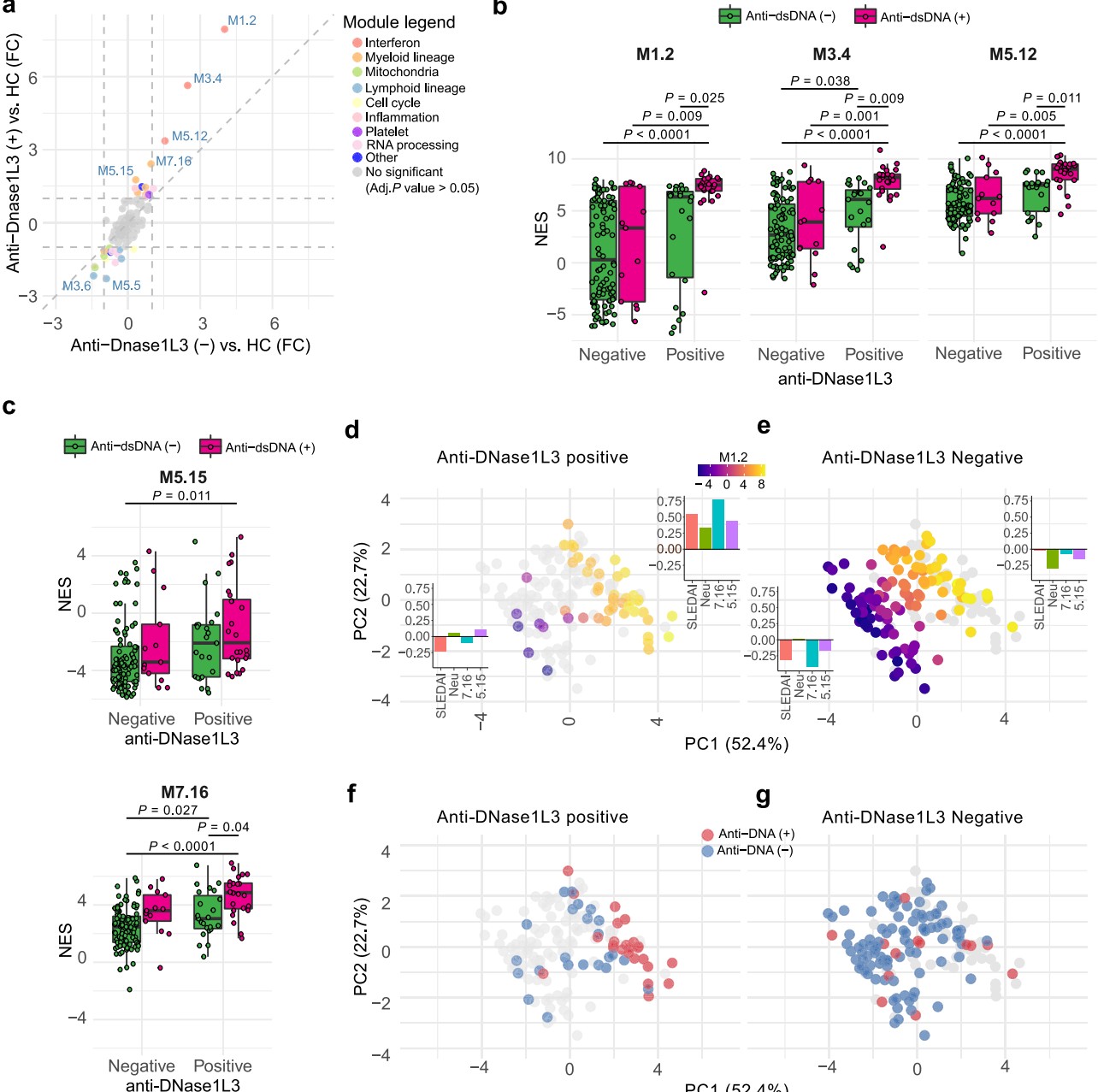

**Fig. 3 | Transcriptional fingerprints associated with anti-DNase1L3 antibodies in SLE. a** Four-way plot of the differentially regulated blood expression modules between anti-DNase1L3 (+) SLE *vs.* healthy controls (HC) (data from Supplementary Table 2), and anti-DNase1L3 (−) SLE *vs.* HC (data from Supplementary Table 3). FC, Fold change. **b**, **c** Comparison of the ssGSEA normalized enrichment scores (NES) for the IFN modules M1.2, M3.4, and M5.12 (**b**), and the myeloid lineage modules M5.15 and M7.16 (**c**) in SLE patients according to anti-DNase1L3 antibody positivity. Variables were compared using two-way ANOVA with Tukey's test as post-hoc. **d**, **e** PCA projection of the significantly regulated modules (M1.2, M3.4, M5.2, M5.15,

and M7.16), neutrophil count, and disease activity in SLE subjects according to anti-DNase1L3 antibody positive (**d**) or anti-DNase1L3 negative (**e**) status. Color scale represents the activity of the IFN module M1.2 (ssGSEA NES score). Bar graphs represent the mean Z score of SLEDAI, neutrophil count (Neu) and activity of the modules M5.15 and M7.16 of patients according to high or low M1.2 activity, defined as patients with PC1 > 0 or PC1 ≤ 0, respectively. **f** Distribution of anti-DNA antibody positive or negative status of anti-DNase1L3 positive SLE patients in PCA projection from **d**. **g** Distribution of anti-DNA antibody positive or negative status of anti-DNase1L3 negative SLE patients in PCA projection from **e**.

(Fig. 4a), demonstrating that a subset of anti-DNase1L3 antibodies derive from autoreactive $V_H$4-34-expressing B cells, and that the proportion of these autoantibodies importantly vary among SLE patients.

To gain further insights into the origin and pathogenicity of anti-DNase1L3 antibodies in SLE, we screened for anti-DNase1L3 antibodies in a set of monoclonal antibodies previously generated from single B cells and antibody-secreting cells (ASCs) from SLE patients experiencing flares[6,7]. In particular, we initially focused on the analysis of 87 monoclonal antibodies largely generated from 9G4+ isotype-switched

memory (SwM) B cells and ASCs[6,7] (Supplementary Fig. 2 and Supplementary Table 11). Among these monoclonals, we found four antibodies with reactivity to DNase1L3 (i.e., 75G12, 75A11, 88F7, and 627A11), which were all derived from SwM B cells (Supplementary Fig. 2). All antibodies were encoded by the self-reactive $V_H$4-34 heavy chain variable region gene (Fig. 4b). Based on the IgH V-D-J usage, complementarity-determining region 3 (CDR3) sequence, CDR3 length and the presence of common mutations, monoclonals 75G12 and 75A11 (isolated from the same patient) were determined to be clonally

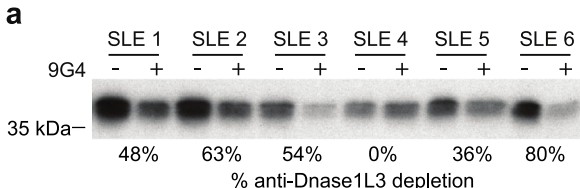

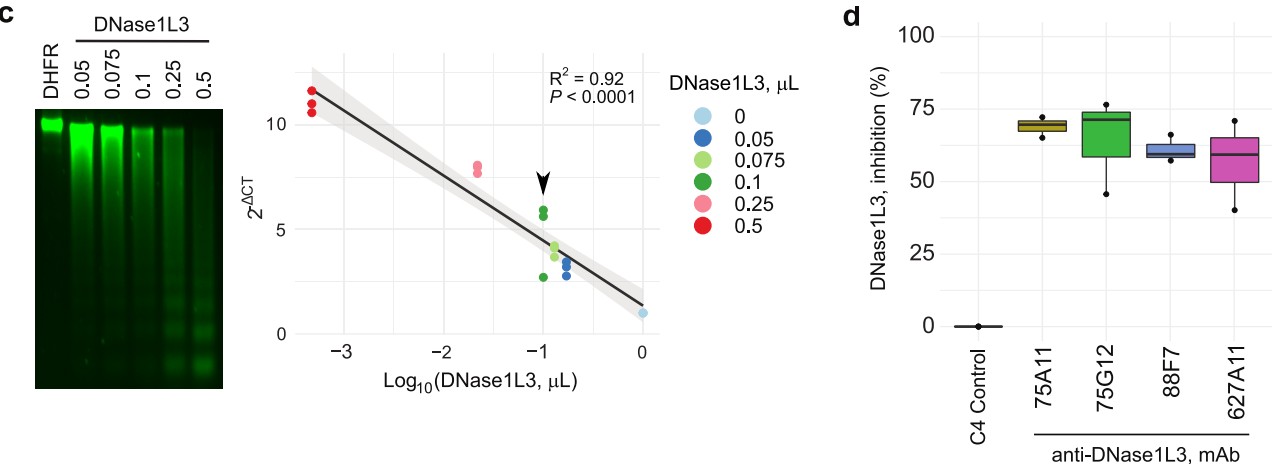

**Fig. 4 | Origin and functional characterization of SLE patient-derived monoclonal antibodies to DNase1L3. a** Radiolabeled DNase1L3 was immunoprecipitated (IP) with anti-DNase1L3 positive SLE sera with (+) and without (−) immunoglobulin (Ig) depletion using 9G4 monoclonal antibodies. Radiolabeled DNase1L3 in immune complexes was quantified by densitometry and anti-DNase1L3 antibody depletion was expressed as percentage according to their corresponding 9G4 non-depleted serum. **b** Ig gene usage, mutation number and CDR3 amino acid sequences of monoclonal antibodies to DNase1L3. **c** Increasing amounts of recombinant DNase1L3 (PURExpress) were incubated with purified intact nuclei. Chromatin degradation was visualized in a 1.5% agarose gel (left panel) and quantified by LM-qPCR (right panel). Recombinant dihydrofolate reductase (DHFR, PURExpress) (1 μL) was used as negative control. The arrowhead indicates the concentration of DNase1L3 used for chromatin degradation assays in **d. d** Effect of monoclonal antibodies (C4 control and anti-DNase1L3) on DNase1L3 activity. Chromatin degradation was quantified by LM-qPCR. The percentage of DNase1L3 inhibition was calculated as $DNase1L3_{\%inh} = 1 - 2^{-\triangle Ct} \times 100$, using the $2^{-\triangle CT}$ from the conditions with DNase1L3 mAb with the C4 control as reference. Experiments were performed in two (**d**) and three (**c**) separate occasions.

related (Fig. 4b). The enrichment of somatic hypermutations (SHM) in all antibodies supports that they originated from antigen-experienced cells.

The function of SLE patient-derived anti-DNase1L3 monoclonal antibodies was further addressed using recombinant DNase1L3 in chromatin digestion assays. Since DNase1L3 catalyzes DNA hydrolysis to produce broken ends that are blunt and 5'-phosphorylated[47], DNase1L3 activity was determined by absolute quantitation of internucleosomally fragmented genomic DNA using ligation-mediated qPCR (LM-qPCR)[48,49]. First, we validated this assay by using increasing amounts of enzyme to digest native chromatin in intact nuclei (Fig. 4c). Since DNase1L3 was generated using a cell-free transcription/translation system, dihydrofolate reductase (DHFR) expressed under similar conditions was used as negative control (Fig. 4c). Then, using an enzyme concentration within the linear range of DNase1L3 activity, chromatin degradation was determined in the presence of monoclonal antibodies to DNase1L3. As a negative control, we used a monoclonal

antibody generated from a single plasmablast (named C4) isolated from a healthy donor (Supplementary Table 12). While the control monoclonal antibody had no effect on chromatin degradation mediated by DNase1L3, the monoclonal antibodies to DNase1L3 decreased DNA hydrolysis by the endonuclease by 60−70% (Fig. 4d).

## $V_H4$-34-derived antibodies to DNase1L3 are also reactive to dsDNA and some to cardiolipin/B2GPI

During the analysis of monoclonal antibodies to DNase1L3, we were surprised that three of these antibodies were initially described as reactive to dsDNA and cardiolipin (i.e. 75G12, 75A11, and 88F7)[6]. To better understand the specificity of these antibodies, we determined the $EC_{50}$ of monoclonals 75G12, 75A11, 88F7, and 627A11 against DNase1L3, dsDNA, and cardiolipin bound to B2GPI (hereafter termed "cardiolipin") (Fig. 5a−c and Supplementary Table 12). Interestingly, the monoclonals showed different patterns of antigen binding (Fig. 5d). Among the clonally related antibodies, 75G12 displayed

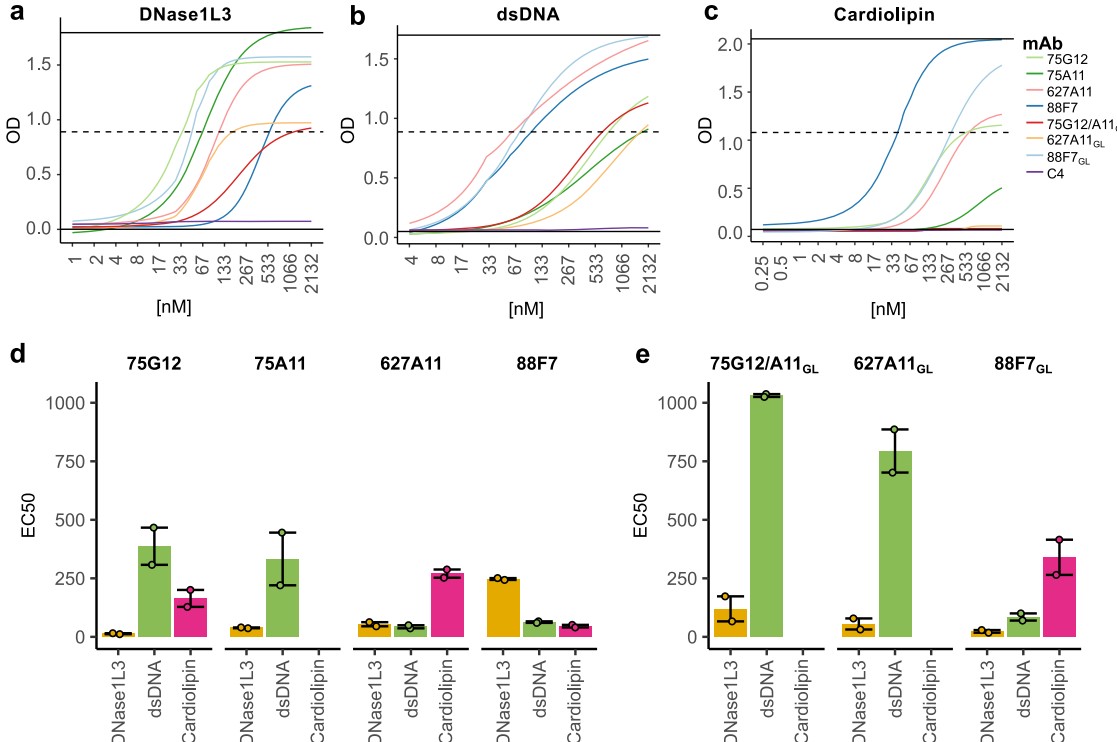

**Fig. 5 | Half-maximal effective concentration (EC$_{50}$) for V$_H$4-34 anti-DNase1L3 monoclonal antibody binding to DNase1L3, dsDNA and cardiolipin.**
**a–c** Monoclonal anti-DNase1L3 antibodies and their germline (GL) variants were titrated against DNase1L3 (**a**), dsDNA (**b**) and cardiolipin (**c**). Curves on the graph correspond to the fitted four-parameter logistic (4PL) model used to calculate the

EC$_{50}$. **d, e** Comparison of the EC$_{50}$ for antibody binding to DNase1L3 (yellow), dsDNA (green), and cardiolipin (pink) using mutated (75G12, 75A11, 627A11 and 88F7) and GL reverted (75G12/A11$_{GL}$, 627A11$_{GL}$ and 88F7$_{GL}$) monoclonal antibodies. Antibodies 75G12 and 75A11 are the same antibody when reverted to GL. Bars show the mean EC$_{50}$ and the error bars represent upper and lower EC$_{50}$ values.

strong preference against DNase1L3 followed by cardiolipin and dsDNA, while binding of 75A11 was more prominent to DNase1L3 followed by dsDNA with no reactivity to cardiolipin (Fig. 5d). Antibody 627A11 showed almost identical reactivity to DNase1L3 and dsDNA with lower binding efficiency to cardiolipin. In contrast, 88F7 was the only antibody in which the binding efficiency was lesser against DNase1L3 compared to dsDNA and cardiolipin (Fig. 5d).

To further understand the autoreactive origin of the monoclonal antibodies, we reverted the heavy chain (IgH) and corresponding light chain (IgL) variable gene sequences to their germline form, and binding to DNase1L3, dsDNA, and cardiolipin were assessed by EC$_{50}$ (Fig. 5a–c, e, and Supplementary Table 13). Importantly, monoclonals 75G12 and 75A11 are the same antibody when reverted to germline (referred to as 75G12/A11$_{GL}$). Strikingly, while binding to DNase1L3 was either unchanged (i.e. clone 627A11$_{GL}$), slightly decreased (i.e. clone 75G12/A11$_{GL}$) or even enhanced (i.e. cloned 88F7$_{GL}$) after antibody reversion to germline, the reactivity of these antibodies to dsDNA or cardiolipin was importantly decreased or completely lost (Fig. 5a–c, e, and Supplementary Table 13). Together, the data suggest that these antibodies primarily arise from autoreactive B cell precursors with higher affinity to DNase1L3, which expanded their binding efficiency to dsDNA and cardiolipin as result of SHM.

**A subset of SLE-derived monoclonal antibodies formerly defined as anti-dsDNA have dual reactivity with DNase1L3**
To address whether dual reactivity to DNase1L3 and dsDNA is exclusive of autoantibodies derived from autoreactive V$_H$4-34$^+$ B cells or is a common feature among anti-dsDNA antibodies, we additionally analyzed four SLE-derived non-V$_H$4-34 IgG monoclonal antibodies originally defined as anti-dsDNA from which both the IgH and IgL variable gene sequences are publicly available (i.e., 32.B9, 33.H11, 33.C9, and RH-14)[50,51]. Their V(D)J usage, CDR3 sequence, and number of SHM are

summarized in Fig. 6a. Importantly, while these antibodies were derived from patients with lupus nephritis and bind to dsDNA with high affinity, they have different features, which are interesting in the setting of anti-dsDNA antibody heterogeneity. Monoclonals 33.C9 and RH-14 are both nephritogenic and deposit in the glomerulus in mice[51,52]. In contrast, monoclonal 32.B9 is not nephritogenic in mice[52]. Antibody 33.H11 has not been tested in vivo. Interestingly, however, this antibody and the nephritogenic monoclonal RH-14 are public clonotypes (Fig. 6a)−i.e. they were isolated from different donors but share the same IgH and IgL V(D)J usage and CDR3 amino acid sequences[53,54]−, suggesting that they may have similar pathogenic potential. Monoclonal RH-14 is cross-reactive with α-actinin[19], and only antibody 33.C9 has been reverted to germline and shown that binding to dsDNA is fully dependent of SHM[55].

Analogous to the V$_H$4-34 monoclonals, we determined the EC$_{50}$ of mutated and germline reverted antibodies 32.B9, 33.H11, 33.C9, and RH-14 against DNase1L3, dsDNA and cardiolipin (Fig. 6b–e and Supplementary Table 14). Importantly, antibodies 33.H11 and RH-14 are the same when reverted to germline (defined as 33.H11/RH-14$_{GL}$). As previously described, we confirmed that the four monoclonal antibodies bind dsDNA with high and similar efficiency (Fig. 6e and Supplementary Table 14). Coincidentally, however, only the nephritogenic monoclonals 33.C9 and RH-14, and antibody 33.H11−which is clonally related to RH-14−showed binding to DNase1L3. Antibody 32.B9, which is considered not pathogenic in mice, had no reactivity to DNase1L3 (Fig. 6e). Different to V$_H$4-34-derived antibodies (Fig. 5d), none of these antibodies showed reactivity with cardiolipin (Fig. 6e). Interestingly, binding to dsDNA by antibody 32.B9 is germline encoded, which is prominently enhanced by affinity maturation (Fig. 6e). In contrast and different to the V$_H$4-34-derived antibodies, dual reactivity to dsDNA and DNase1L3 by antibodies 33.C9, RH-14, and 33.H11 is completely dependent on SHM (Fig. 6e). Consistent with their reactivity to

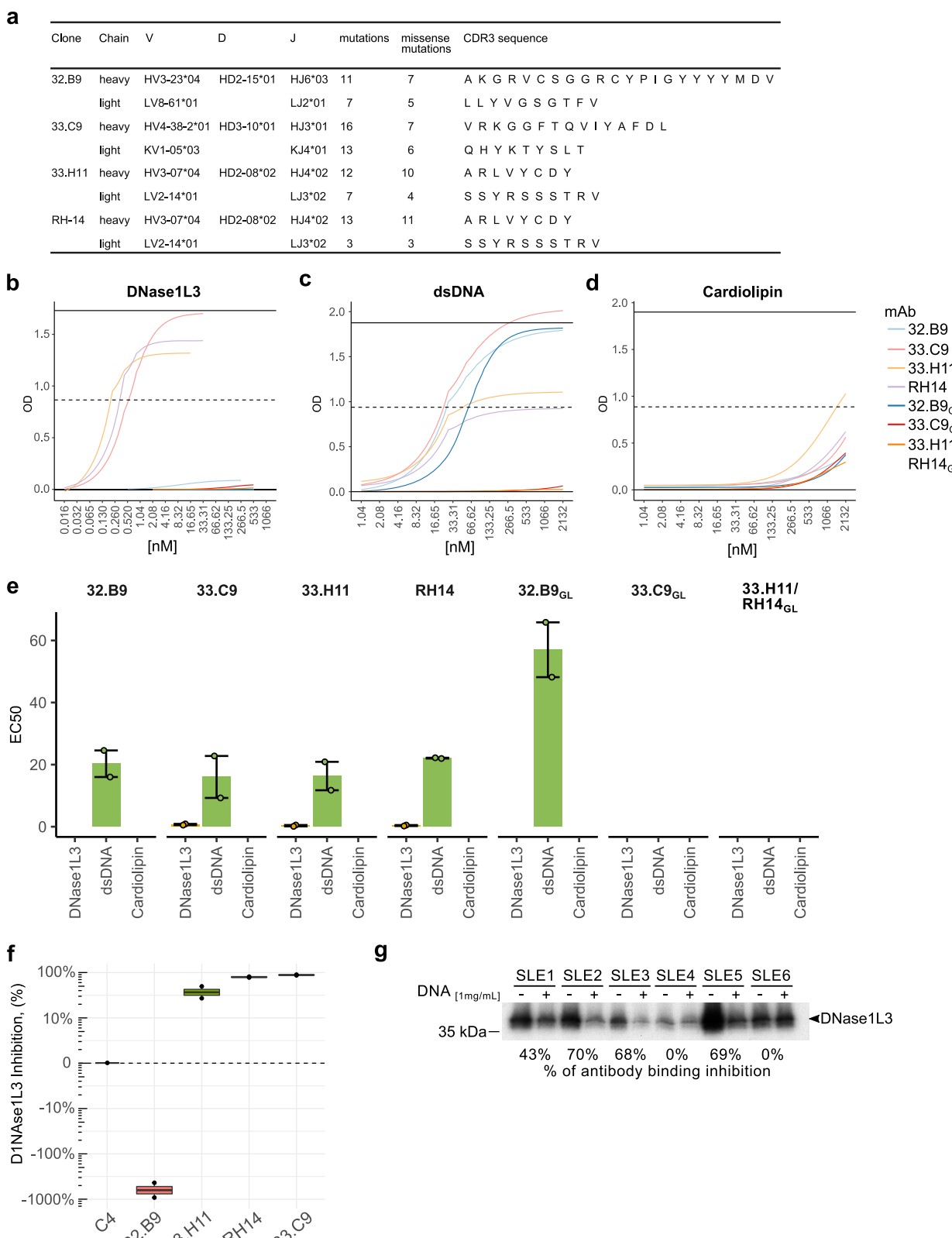

DNase1L3, monoclonals 33.C9, RH-14, and 33.H11 inhibited the activity of the enzyme, although 33.H11 was the least efficient of these antibodies (Fig. 6f). Unexpectedly, we found that the presence of antibody 32.B9, which has no reactivity to DNase1L3, importantly enhanced chromatin degradation (Fig. 6f). Since 32.B9 binds to dsDNA, it is possible that the antibody may increase dsDNA degradation by facilitating accessibility to the substrate.

Lastly, to demonstrate that dual reactivity to DNase1L3 and dsDNA is not exclusive for monoclonal antibodies, we confirmed the existence of these autoantibodies in SLE serum by competing anti-DNase1L3 antibody binding against dsDNA. In the presence of dsDNA, serum reactivity against DNase1L3 was variable affecting from 0% to 70% antibody binding (Fig. 6g), which is consistent with clinical data suggesting the presence of autoantibodies with

**Fig. 6 | Characterization of pathogenic and non-pathogenic SLE-derived monoclonal antibodies categorized as anti-dsDNA. a** Ig gene usage, mutation number and CDR3 amino acid sequences of monoclonal antibodies 32.B9, 33.H11, 33.C9, and RH-14. **b**–**d** Mutated and germline reverted (GL) antibodies 32.B9, 33.H11, 33.C9, and RH-14 were titrated against DNase1L3 (**b**), dsDNA (**c**) and cardiolipin (**d**). Curves on the graph correspond to the fitted four-parameter logistic (4PL) model used to calculate the $EC_{50}$. **e** Comparison of the $EC_{50}$ for antibody binding to DNase1L3 (yellow), dsDNA (green), and cardiolipin (pink) using mutated (32.B9, 33.H11, 33.C9, and RH-14) and GL reverted (32.B9$_{GL}$, 33.C9$_{GL}$ and 33.H11/RH14$_{GL}$) monoclonal antibodies. Monoclonals 33.H11 and RH-14 are the same antibody when reverted to GL. Bars show the mean $EC_{50}$ and the error bars represent upper and lower $EC_{50}$ values. **f** Effect of monoclonal antibodies (C4 control, 32.B9, 33.H11, 33.C9, and RH-14) on DNase1L3 activity. Chromatin degradation was quantified by LM-qPCR as described in Fig. 4d. Experiments were performed in two separate occasions (**b**–**d**, **f**). **g** Radiolabeled DNase1L3 was immunoprecipitated (IP) in the absence (−) or presence (+) of salmon sperm DNA (1 mg/mL) using representative SLE sera positive for antibodies to DNase1L3. Relative quantification of IP inhibition was calculated by the ratio of the densitometric value of sera with/without DNA.

mono and dual specificity to dsDNA and DNase1L3 in patients with SLE.

## Discussion

While serological detection of autoantibodies is a useful tool for diagnosis in SLE, these assays cannot discriminate whether binding to specific antigens is due to cross-reactivity or by the existence of monospecific autoantibodies. This caveat likely explains why clinical and pathogenic associations with autoantibodies thought to be antigen-specific are heterogeneous in SLE. Here, we showed that a subset of autoantibodies that would be categorized as being monospecific for either dsDNA or DNase1L3, depending on the detection assay used, are actually autoantibodies with reactivity to both antigens. In addition to the clinical and diagnostic implications of these findings, these data provide novel insights into the origin and mechanisms of pathogenic anti-dsDNA antibodies in SLE.

While we found a striking association between anti-DNase1L3 and anti-dsDNA antibodies, which is explained in part by the presence of double reactive antibodies to DNase1L3 and dsDNA, it is puzzling that a recent study by Hartl et al.[21] did not seem to support this finding, even though both studies found that antibodies to DNase1L3 were significantly linked to renal disease in SLE. Indeed, the only association described by Hartl et al. with anti-DNase1L3 antibodies was renal involvement, which is likely explained by the fact that their cohort was highly enriched in patients with renal disease (87/120, 72%) and antibodies to DNase1L3 were only determined in a subset of these patients (57/120, 48%)[21], limiting their capacity to find other clinical and serological associations described in our study. Interestingly, the study by Hartl et al. neither found an association between anti-DNase1L3 and anti-C1q antibodies[21], which are highly prevalent in patients with renal involvement. Considering that antibodies to dsDNA and C1q are present in up to 80% and 100% of patients with renal disease, respectively[56], and that 72% of the patients had renal involvement, it is surprising that Hartl et al. found no associations of these antibodies with anti-DNase1L3 antibodies, unless a significant number of patients with renal disease were seronegative for antibodies to dsDNA and C1q. Alternatively, Hartl et al. only examined the correlation between autoantibody levels, which was not significant, rather than the association with antibody positivity, as we did in our study. For the correlation analysis, Hartl et al. only included small subsets of patients in which the autoantibodies were detected using a homemade bead-based antigen array [i.e. anti-dsDNA in 33/120 (28%) and anti-C1q in 25/120 (21%) patients]. Aside from the small sample size, finding a significant correlation between antibody levels is unlikely because cross-reactive anti-DNase1L3/dsDNA antibodies only correspond to a subset of antibodies detected by the independent anti-DNase1L3 and anti-dsDNA assays.

The analysis of SLE-derived monoclonal antibodies identified two subsets of anti-DNase1L3/dsDNA antibodies according to their origin. A subset uses the variable heavy-chain gene segment $V_H4$-34, which encodes for a significant number of autoantibodies in active SLE. $V_H4$-34$^+$ B cells, bearing the idiotype 9G4, are inherently autoreactive cells largely excluded from the germinal centers and underrepresented in the memory compartment in healthy individuals[57]. In patients with SLE, however, 9G4-B cells progress through this checkpoint and successfully participate in germinal center reactions, generating increased levels of IgG memory and plasma cells[58]. 9G4 antibodies represent 10–45% of the total serum IgG in patients with active disease[42–44,58], which account for the vast majority of anti-B cell CD45 antibodies and a significant fraction of anti-dsDNA in SLE.

Within a large set of 9G4 SLE-derived monoclonal antibodies, we identified 4 antibodies reactive to DNase1L3, which also bind dsDNA and some to cardiolipin. Further analysis of these antibodies showed that while binding to dsDNA or cardiolipin was decreased in their germline form, binding to DNase1L3 was minimally affected or even enhanced, suggesting that these antibodies originated from autoreactive B cell precursors with preferential reactivity against this endonuclease. The finding that SHM increased the binding of antibody clones 75G12, 75A11, and 627A11 to dsDNA and of clones 75G12 and 627A11 to cardiolipin implicates SHM as the driver of cross-reactivity in these autoantibodies. This notion is further supported by the finding that antibodies 75G12 and 75A11 are clonally related with a single amino acid difference in HCDR2[6], which is responsible for their differential binding to cardiolipin. Monoclonal 88F7 is particularly interesting, because it binds DNase1L3 more efficiently in its germline form, suggesting that affinity maturation of this antibody may not have been directly driven by DNase1L3. Since DNase1L3 interacts with dsDNA, it is possible that dsDNA and DNase1L3 are both involved in the process of affinity maturation of this subset of anti-DNase1L3 antibodies, facilitating the production of dual-reactive autoantibodies in which the affinity against either antigen may be stochastically determined.

The second subset of anti-DNase1L3/dsDNA antibodies, originally defined as nephritogenic antibodies to dsDNA, are encoded by diverse $V_H$ gene segments. Binding efficiency of this set of antibodies is higher to DNase1L3 than dsDNA, suggesting that DNase1L3 is the primary target and dsDNA is the cross-reactive antigen. Nevertheless, like the $V_H4$-34-derived subset, we cannot exclude the possibility that both DNase1L3 and dsDNA might be involved in the affinity maturation of this set of autoantibodies. In contrast to the $V_H4$-34-derived subset, however, binding of these antibodies to dsDNA and DNase1L3 is entirely dependent on SHM and none of the antibodies showed reactivity to cardiolipin. Together, these findings imply that anti-DNase1L3/dsDNA antibodies are heterogenous regarding their origin, reactivities and can be generated from both autoreactive and non-autoreactive B cell precursors. Moreover, the data demonstrate that these antibodies can be incorrectly categorized as being monospecific for either DNase1L3 or dsDNA— and some to cardiolipin—when tested by a single assay.

Independently of their origin, the presence of autoantibodies with binding capacity to both DNase1L3 and dsDNA has important implications for SLE pathogenesis. By targeting distinct autoimmune pathways in parallel, these antibodies may have the ability to amplify their pathogenic potential. For instance, these antibodies can increase the load of extracellular dsDNA by blocking DNase1L3 activity, while binding to dsDNA can generate immune complexes promoting IFN-I production and cause direct damage by depositing in tissues, such as the kidney. In this regard, it is noteworthy that SLE patients double positive for anti-DNase1L3 and anti-dsDNA antibodies showed the

most striking association with disease activity and the IFN and myeloid signatures when compared with single positive patients for either antibody specificity, or negative for both autoantibodies. Thus, the coexistence of serum reactivities to dsDNA and DNase1L3 seems to identify a subset of pathogenic antibodies linked to higher disease activity in SLE. In the context of the analysis of monoclonal antibodies, the most rational explanation is that these pathogenic antibodies correspond to the subset with dual reactivity to DNase1L3 and dsDNA.

The finding that some monoclonal antibodies also cross-react with cardiolipin is intriguing, particularly because these antibodies target cardiolipin bound to B2GPI and therefore have the potential to be pathogenic. Although this study did not focus on the analysis of anti-phospholipid syndrome, it is worth noting that anti-DNase1L3 antibody positivity was significantly associated with livedo, lupus anticoagulant, antibodies to cardiolipin and B2GPI, and marginally with arterial thrombosis. Thus, it is possible that triple reactive antibodies−targeting DNase1L3, dsDNA and cardiolipin−may contribute to the pool of anti-phospholipid antibodies in SLE.

The study of SLE serum and monoclonal antibodies also demonstrated that not every antibody catalogued as anti-dsDNA or anti-DNase1L3 has dual reactivity to these antigens. There are a significant number of SLE sera that are only positive for anti-dsDNA or anti-DNase1L3 antibodies, not all anti-DNase1L3 antibodies in serum were blocked by dsDNA, and we identified one anti-dsDNA monoclonal antibody (32.B9) with no reactivity to DNase1L3. Monoclonal 32.B9 is particularly interesting because binding to dsDNA is germline encoded, reactivity to dsDNA is enhanced after SHM, and despite having the same binding efficiency to dsDNA as the dual reactive antibodies 33.C9 and RH-14, it is not nephritogenic in mice. Moreover, in contrast to dual reactive antibodies that block DNase1L3 activity, we were surprised that monoclonal 32.B9 enhanced chromatin degradation by DNase1L3, highlighting the incredible heterogeneity in the function of antibodies catalogued as anti-dsDNA in SLE. Whether antibody 32.B9 and others with similar activity may instead be protective for SLE by promoting DNase1L3-mediated chromatin degradation is a hypothesis that will need further exploration. Moreover, these findings underscore that studying the activity of anti-DNase1L3 antibodies in serum or using bulk IgG from SLE patients should be avoided because it will be affected by the presence of different autoantibodies targeting components in the substrate, such as anti-dsDNA antibodies.

Antibody 32.B9 was also useful to demonstrate that binding to DNase1L3 is not a general feature of anti-dsDNA antibodies. Instead, we propose that anti-DNase1L3/dsDNA antibodies correspond to a distinct subset of autoantibodies in which the primary substrate is DNase1L3. This notion is particularly appealing, as mammalian dsDNA is poorly immunogenic[59] and the identity of the antigen that elicits the production of antibodies to dsDNA remains unclear. Indeed, the previous finding that a peptide surrogate for dsDNA can induce anti-dsDNA antibodies and nephritis in mice supports the notion that a protein antigen can trigger the induction of antibodies reacting with dsDNA[60]. The discovery of anti-DNase1L3/dsDNA antibodies renews interest in better understanding the significance of cross-reactivity among autoantibodies in SLE, which may shed light on the origin and heterogeneity among the wide range of autoantibodies found in this autoimmune disease.

# Methods

## Study design
The objective of this study was to investigate the origin and immunopathology related to anti-DNase1L3 antibodies in patients with SLE. To evaluate this, sera from 62 healthy controls and 158 SLE patients from the "Study of biological Pathways, Disease Activity and Response markers in patients with Systemic Lupus Erythematosus" (SPARE)[61] cohort were studied to define the prevalence and clinical significance of antibodies to DNase1L3 in SLE. SPARE is a prospective observational

cohort that has been extensively described previously[33,61]. Briefly, adult patients (age 18 to 75 years-old) who met the definition of SLE per the revised American College of Rheumatology classification criteria were eligible into the study[62]. Sex and gender of each participant was collected based on self-report. At baseline, the patient's medical history was reviewed, and information on current medications was recorded. Patients were followed-up over a 2-year period. Patients were treated according to standard clinical practice. Disease activity was assessed using the Safety of Estrogens in Lupus Erythematosus: National Assessment (SELENA) version of the Systemic Lupus Erythematosus Disease Activity Index (SLEDAI)[63] and physician global assessment (PGA)[64]. C3, C4, anti-dsDNA (Crithidia), complete blood cell count and urinalysis were performed at every visit. Study participants also underwent whole blood gene expression analysis at baseline using the Affymetrix GeneChip HT HG-U133+[33,61]. Eighty-seven monoclonal antibodies previously generated from SLE patients experiencing flares were used to screen for anti-DNase1L3 antibodies[6,7]. All samples were obtained with written informed consent from the participants. The study protocol was approved the Institutional Review Boards at the Johns Hopkins University School of Medicine and Emory University School of Medicine.

## DNase1L3 cloning and protein expression
cDNA encoding human mature DNase1L3 (amino acids 21-305) was synthesized using RNA from human PBMCs and cloned into pcDNA3.1 and pET-28a(+). In pET-28a, the 5' site end in mature DNase1L3 was cloned at the NcoI site in the vector. Thus, the T7 promoter is followed by a ribosome binding site and the DNase1L3 start codon. The protein encoded by pET-28a-DNase1L3 is not tagged. pcDNA3.1-DNase1L3 was used to generate $[^{35S}]$methionine-labeled DNase1L3 by TNT T7 Quick Coupled Transcription/Translation (Promega). pET-28a-DNase1L3 was used to generate recombinant active DNase1L3 using the PURExpress In Vitro Protein Synthesis Kit (New England Biolabs). The control plasmid encoding dihydrofolate reductase (DHFR) is included in the PURExpress kit.

## Detection of antibodies to DNase1L3
Radiolabeled DNase1L3 was immunoprecipitated with 2 µl of serum, 10 µL of cell supernatants from monoclonal antibody producing cells, or with purified monoclonal antibodies in 300 µL of NP-40 buffer (20 mM Tris/HCl, 150 mM NaCl, 1 mM EDTA, 1% Nonidet P40, pH 7.4) for 1 hr at 4 °C. In some experiments, anti-DNase1L3 binding was performed in the presence of 1 mg/mL Salmon Sperm DNA (Thermo Fisher Scientific). Protein A beads were added and incubated for additional 30 min at 4 °C. After three washes with vortexing in NP-40 lysis buffer, beads were boiled in SDS sample buffer. Samples were separated by gel electrophoresis, and immunoprecipitated proteins were visualized by radiography. Densitometry was performed on all films and values were normalized to a high-titer anti-DNaseL13 serum. Antibody positivity was defined using a cutoff of two standard deviations above the mean anti-DNase1L3 antibody level in healthy sera.

## Cloning of monoclonal antibodies
The cloning of monoclonal antibodies from single B cells and antibody-secreting cells (ASCs) from SLE patients experiencing flares was previously described[6,7]. Single peripheral blood plasmablasts (CD3-CD14-CD19 + CD20-CD27 + CD38+) from a healthy female donor were sorted into 96-well PCR plates. IgH and IgL (κ or λ) variable regions were cloned into expression vectors containing human Igγ1, Igκ, or Igλ constant regions as previously described[65]. One monoclonal antibody (C4) was highly expressed and was selected for further characterization. Analysis of the antigen specificity of C4 was performed using the HuProt human proteome microarray (CDI). C4 exhibited broad, non-specific reactivity to 0.25% of the >21,000 proteins on the array, but had no reactivity to DNAse1L3. This antibody

was used a monoclonal control. The IgH and IgL variable gene sequences of monoclonal antibodies 32.B9, 33.H11, 33.C9, and RH-14[50,51] were synthetized using the Custom gene synthesis service from Integrated DNA Technologies (IDT), and cloned into expression vectors containing human Igγ1, Igκ or Igλ constant regions (kindly provided by Eric Meffre, Yale University School of Medicine, New Haven, CT).

## Reversion of SHM sequences to germline

The V(D)J germline sequences with the lowest number of mismatch nucleotides compared to mutated sequences were obtained using IgBLAST, synthetized using the Custom gene synthesis service from IDT, and cloned into expression vectors containing human Igγ1, Igκ or Igλ constant regions.

## Monoclonal antibody production

Monoclonal antibodies were produced using 293T cells in high glucose DMEM and 10% ultra-low IgG fetal bovine serum (Gibco), Expi293 cells and ExpiCHO cells (Thermo Fisher Scientific) by co-transfecting plasmids encoding IgH and IgL according to the manufacturer instructions. Supernatants were collected at day 5 after transfection and the antibodies purified using Protein A beads (Pierce). We found no differences among monoclonal antibodies produced by either cell type, although ExpiCHO cells are certainly the most efficient to produce larger amounts of antibodies.

## Quantitation of inter-nucleosomally fragmented genomic DNA using ligation-mediated qPCR (LM-qPCR)

DNA degradation was measured by quantitation of inter-nucleosomally fragmented DNA by LM-qPCR as previously described[49] with some modifications. Briefly, the DHApo1 and DHApo2 oligonucleotides (5′ to 3′: AGCACTCTCGAGCCTCTCACCGCA and TGCGGTGAGAGG, respectively) were annealed by mixing 50 μL (100 pmol/μl) of each in 250 μl of 250 mM Tris (pH 7.7), heating the mixture to 90 °C for 5 min, incubating at 55 °C for 15 min, and allowing the mixture to cool to RT. The linker mixture was frozen and thawed on ice before use. Ligation reactions (20 μL) were performed overnight at 16 °C using Quick T4 ligase (New England Biolabs), 100 ng DNA and 1 μL linker. qPCR was performed using the DHApo1 primer and iTaq Universal SYBR Green Supermix (Bio Rad) as follow: 95 °C for 4 min (1 cycle), 72 °C for 4 min (1 cycle), followed by 40 cycles of denaturation at 94 °C for 1 min and annealing/extension at 72 °C for 3 min.

## Neutralizing activity of monoclonal antibodies to DNase1L3

DNase1L3 and control DHFR were generated using the PURExpress In Vitro Protein Synthesis Kit (NEB). DNase1L3 activity was titrated by co-incubating increasing amounts of PURExpress DNase1L3 (0.05 μL, 0.075, 0.1 μL, 0.25 and 0.5 μL) with 10,000 purified nuclei in 20 μL of DNase reaction buffer (10 mM Tris/HCl, 50 mM NaCl, 2 mM MgCl$_2$, 2 mM CaCl$_2$, pH 7.0) containing 5% bovine serum albumin (BSA) solution (Sigma). PURExpress DHFR was used as negative control. After 30 min at 37 °C, reactions were stopped by adding 20 μL of proteinase K buffer (PKB, 50 mM Tris-HCL pH 8.0, 1 mM EDTA, 2.5% Tween 20, and 800 Units/mL proteinase K), incubation at 50 °C for 20 min and heat inactivation at 95 °C for 5 min. DNA was purified by isopropanol precipitation and resolved in a 1.5% agarose gel. In addition, DNA degradation was measured by quantitation of inter-nucleosomally fragmented DNA by LM-qPCR. The amount of inter-nucleosomally fragmented DNA was calculated by the $2^{-\triangle Ct}$ method, using the condition with PURExpress DHFR as reference. No DNase activity was found in the PURExpress synthesis kit unless DNase1L3 was expressed (Fig. 4c). To assess the effect of anti-DNase1L3 monoclonal antibodies on DNase1L3 activity, antibodies at 1.6 μM were incubated with 0.1 μL of PURExpress DNase1L3 in DNase reaction buffer containing 5% BSA. The monoclonal antibody C4 was used as antibody control. In addition, purified nuclei were incubated with PURExpress DHFR as control for undigested DNA. After 1 hr at room temperature (RT), 10,000 purified nuclei were added and further incubated for 30 min at 37 °C. The final volume of the reaction was 20 μL. Reactions were stopped with PKB as described above and DNA was purified by isopropanol precipitation. Inter-nucleosomally fragmented DNA was quantified by LM-qPCR. The inter-nucleosomally fragmented DNA was calculated by the $2^{-\triangle Ct}$ method with C4 control as reference. The percentage of DNase1L3 inhibition was calculated as $DNase1L3_{\%inh} = 1 - 2^{-\triangle Ct} \times 100$.

## Serum depletion of antibodies bearing the 9G4 idiotype

Sera depletion was performed using the Pierce™ Protein G IgG Plus Orientation Kit (Thermo Fisher Scientific) primarily to manufacture suggestions. The idiotypic rat anti-human 9G4 mAb was saturated within the agarose and cross-linked utilizing a multi-flow through load. 500 μL of patient sera was utilized and bound overnight at 4 °C prior to flow through. The elution protocol and column wash resulted in roughly a 1:5 dilution of the sera volume following flow through. Total IgG and 9G4 specific ELISAs were performed on the sera pre- and post-depletion. The total IgG loss ranged from 10 to 27% following the 9G4 column depletion, however the 9G4 IgG loss was >99% in all samples.

## Half maximal effective concentration (EC$_{50}$) assays

Anti-DNase1L3 monoclonal antibodies and their germline variants were titrated in duplicate against dsDNA and cardiolipin bound to B2GPI using QUANTA Lite® dsDNA (Inova Diagnostics, Cat: 708510) and QUANTA lite® ACA IgG III (Inova Diagnostics, Cat: 708625) ELISA assays, respectively. For antibody binding to DNase1L3, we developed an in-house magnetic bead-based immunoassay. Mature DNase1L3 (amino acids 21-305) containing a N-terminal FLAG-tag sequence (FLAG-DNase1L3) was generated by TNT T7 Quick Coupled Transcription/Translation (Promega) and purified using anti-DYKDDDDK Magnetic Agarose beads (Pierce™). FLAG-IRF3 generated by TNT T7 Quick Coupled Transcription/Translation was used as FLAG-tag protein control. FLAG-IRF3 was chosen because the plasmid was already available in the lab. After extensive washes with NP40-buffer, beads containing FLAG-tagged proteins were blocked for one hour with Protein-Free (TBS) Blocking Buffer (Pierce™) and further incubated in duplicate with decreasing concentrations of monoclonal antibodies for one hour using 96-well plates. A 96-well plate magnet was used to keep the beads in the wells during manual washes. After three washes with NP-40 buffer, anti-DNase1L3 antibody binding to FLAG-DNase1L3 or FLAG-IRF3 beads was detected using a horseradish peroxidase (HRP)-conjugated goat anti-human IgG secondary antibody (Jackson Immunoresearch, Code: 109-035-088) diluted 1:10,000 in Protein-Free (TBS) Blocking Buffer (Pierce™). SureBlue TMB peroxidase substrate (KPL) was added after washing the beads with NP-40 buffer to visualize antibody binding and an equal volume of 1 M hydrochloric acid was added to stop the colorimetric reaction, before determining the absorbance at 450 nm. Individual values were corrected for background by subtracting the reactivity to FLAG-IRF3 beads. Relative EC$_{50}$ values were calculated using a 4 parameter logistic regression (4PL) model[66]. The average EC$_{50}$ was determined from two independent experiments.

## Gene expression analyses

Gene expression analysis from the SPARE cohort was previously described[33]. CEL files were subjected to RMA background correction, and quantile normalization, using the Oligo package[34]. To select only expressed genes in whole blood, we filtered out transcripts that had a raw signal <100 in <10% of samples with the genefilter R package. All calculations and analyses were performed using R (ver 4.0.2) and Bioconductor (ver 3.13)[67]. Differentially expressed transcripts (DETs) were analyzed using the R package limma[68]. Functional gene set

enrichment was carried out with the R interface gprofiler2 for the server g:Profiler[69].

## Calculation of the blood expression modules activity

DETs were analyzed using the R package limma[68]. The Blood gene expression modules from Chaussabel et al.[38] were obtained from the R package "tmod" for Bioconductor[70]. Module activity at the individual level was calculated by ssGSEA[37]. Differentially regulated modules according to anti-DNase1L3 status were analyzed with a linear model approach using the R package limma.

## Statistical analyses

Comparisons of continuous variables between groups were done using Student's $T$ test and ANOVA test as indicated. The Mann-Whitney's $U$ test and Kruskal–Wallis test was used for group-wise comparisons of non-normally distributed variables Fisher's exact test and $\chi^2$ tests were used for univariate analysis on SPARE cohort variables, as appropriate. Exact2x2 package in R version 3.5.1 was used for binary variables to obtain $p$-value, OR, and 95% CI. Multivariate analyses were carried out using multivariate logistic regression or multivariate linear regression as indicated. Unsupervised hierarchical clustering with complete linkage was performed by computing a correlation-based distance between genes (Pearson's method) and the Canberra metric for the distance between subjects. Heatmap visualization was done using the Complex heatmap R package[71]. To improve visualization, dendrograms were reordered using the modular leaf ordering methods from the dendsort R package[72]. Statistical significance was set at $p < 0.05$. Since 94% of study the study population were female (Supplementary Table 1), we did not carry out analyses disaggregated for sex and gender. The statistical analyses were carried out with the R software version 4.0.2 and SPSS IBM statistics version. 25.

## Reporting summary

Further information on research design is available in the Nature Portfolio Reporting Summary linked to this article.

## Data availability

Microarray data are available from the Gene Expression Omnibus under accession numbers GSE45291 and GSE121239. Serum from patients with SLE and healthy controls can be obtained under request through material transfer agreements. The raw numbers for charts and graphs are provided in the Source Data file whenever possible. Source data are provided with this paper.

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

## Acknowledgements

We would like to thank the support of all members of the Petri, Sanz, and Andrade labs. Special thanks to Dr. S. Sam Lim and the Grady Memorial Hospital in Atlanta, as well as Emory University Hospital, and the University of Rochester Medical Center for sample support, and all the willing donors from the SPARE Cohort and from all contributing institutions. This project was supported by the Rheumatology Research Foundation (F.A.), the Jerome L. Greene Foundation (F.A. and E.D.), the National Institute of Arthritis and Musculoskeletal and Skin Diseases (NIAMS), and the National Institute of Allergy and Infectious Diseases (NIAID) at the National Institutes of Health (NIH) grants number R01 AR069569, R21 AI147598, R21 AI169851 (F.A.), R01 AR069572 (M.P.), P01 AI125180, R37 AI049660, U19 AI110483 (I.S.), and the Intramural Research Program of the National Institute on Aging to Ranjan Sen (C.A.S.-B.). The contents of this article are solely the responsibility of the authors and do not necessarily represent the official views of the NIH, NIAMS or NIAID. Y.Y. received a scholarship from The Wuhan Scientific Funding for young investigators: 201271031432.

## Author contributions

Conception: F.A. Designing research studies: Y.Y., E.G.-B., J.L., K.S.C., M.I.T.-Z., R.B., Y.W., A.S.C., M.P., I.S., and F.A. Conducting experiments: Y.Y., E.G.-B., K.S.C., R.B., Y.W., A.S.C., M.Pz, D.P.F., M.I.T.-Z., and F.A. Data acquisition and analyses: Y.Y., E.G.-B., J.L., K.S.C., R.B., Y.W., A.S.C., M.Pz, M.I.T.-Z., D.P.F., D.W.G., E.D., M.P., I.S., and F.A. Interpretation of data: Y.Y., E.G.-B., J.L., K.S.C., E.D., M.P., I.S., and F.A. Providing reagents: C.A.S.-B., M.P., E.D., and I.S. Writing—original draft: E.G.-B and F.A. Writing—review and editing: Y.Y., E.G.-B., J.L., K.S.C., R.B., Y.W., A.S.C., M.P., M.I.T.-Z., D.P.F., C.A.S.-B., D.W.G., E.D., M.P., I.S., and FA. All authors reviewed, edited, and approved the manuscript.

## Competing interests

The authors declare no competing interests.
