## [Peer Review File · Nature Communications]

Affinity maturation generates pathogenic antibodies with dual reactivity to DNase1L3 and dsDNA in systemic lupus erythematosusREVIEWER COMMENTS

Reviewer #1 (Remarks to the Author):

Pathogenic anti-dsDNA antibodies arise during affinity maturation of autoantibodies targeting a protein antigen in SLE

Key results

The authors provide original and interesting insights on the multiple reactivity of anti-dsDNA antibodies in patients affected by SLE.

The authors indeed characterize a subgroup of anti-dsDNA capable of inhibiting the activity of circulating DNase, possibly contributing to the pathogenetic role of circulating DNA in increasing the load of immunocomplexes, IFN production and disease activity. Moreover, the authors provide proof that in some cases DNase might represent the first inducer of auto antibodies, and that the cross-reactivity to dsDNA might appear in a second phase, after somatic hypermutations. Moreover, the authors confirm previous reports of cross-reactivity between anti-dsDNA/antiDNase and anti-cardiolipin.

The scientific and clinical relevance of the report was highlighted.

Data are clear and well presented. The experimental procedures are well-structured and clearly conducted. Furthermore, the article is competently constructed, and the analysis was efficiently performed.

No breaches of ethical practice were noted.

Here are some comments for the authors to consider:

Title

I suggest changing the title to specify the target antigen of the autoantibodies characterized in the article. I would consider including the name of anti-DNase1L3, rather than mentioning a "protein antigen".

Abstract

The abstract is well constructed.

The paragraph "A subgroup of these antibodies is encoded by the inherently autoreactive VH4-34 gene segment, derives from anti-DNase1L3 germline-encoded precursors, and gain cross-reactivity to dsDNA after somatic hypermutations. In a second subgroup, originally defined as nephritogenic anti-dsDNA antibodies, dual reactivity to DNase1L3 and dsDNA is gained during affinity maturation, but their binding efficiencies favor DNase1L3 as the primary antigen" would benefit from rewording to better describe the results.

Introduction

- The heterogeneity of anti-dsDNA antibodies should be better elucidated. Previous studies indeed have investigated the different cross-reactivity of these autoantibodies; please consider providing a description of the possible cross-reactive antigens.
- DNase is a limiting factor also for NET accumulation and thus has an important preventive role in the pathogenesis of SLE. I suggest mentioning in the introduction section the role of NETosis in the pathogenesis of the disease with regard to DNase inhibition by auto-antibodies. References should be included.

Results

Fig 1. I suggest modifying the figure title" Prevalence and clinical associations of antibodies to

DNase1L3 in a prospective observational cohort of patients with SLE." including also the serologic associations.

Lines 167-170: the authors define the effect of anti-dsDNA and anti-DNase autoantibodies as "additive" on IFN modules. I suggest reformulating the phrase since the experiments carried out do not explain the direct effect of autoantibodies on IFN modules.

Fig 2.B Please add the text: orange green red over the clusters in order to better understand figure 2D.

Add in the legend what the 3 clusters mean

In addition, I would suggest describing better in the results section and then commenting in the discussion section on the role of increased neutrophil count in anti-DNase1L3 positive versus negative patients. Otherwise, I suggest moving the result to the supplementary section.

Discussion

What could be the clinical use of characterizing the specificity of anti-dsDNA? Would it be possible to identify nephritogenic antibodies to better treat or follow up patients at risk of nephritis?

Do the authors consider their results relevant also from a diagnostic/therapeutic point of view? Could the blockade or the depletion of anti-DNase be a potential different therapeutic target in Lupus or Lupus Nephritis patients?

I suggest better discussing the cross-reactivity with cardiolipin.

Would it have a clinical impact on the patients? Do the patients also have a pro-thrombotic phenotype?

Materials and Methods

Please provide a description of how cardiolipin was tested for anti-cardiolipin reactivity detection. Did the authors use cardiolipin alone or with the protein co-factor (beta-2glycoprotein?)?

Analytical approach

The approach is valid and comprehensive of any statistical tests used for the analysis.

References

Please delete ref 19 as it has not been published yet.

Reviewer #2 (Remarks to the Author):

This is an outstanding original paper, demonstrating for the first time the existence of autoantibodies in SLE that target both dsDNA and the endonuclease DNase1L3. This subset of cross-reacting autoantibodies exhibit stronger pathogenicity. The data suggest that DNase1L3 may be the primary target of autoantibodies that cross-react with dsDNA. The results, which were methodologically well validated, change the perspective on the origin of anti-dsDNA autoantibodies and their pathogenic role in SLE.

Reviewer #3 (Remarks to the Author):

The manuscript by Gomez-Bañuelos et al. entitled "Pathogenic anti-dsDNA antibodies arise during affinity maturation of autoantibodies targeting a protein antigen in SLE" investigates the etiology of autoantibodies targeting DNase1L3 in lupus. In their independent cohort, the authors confirm previous findings that anti-DNase1L3 antibodies are present in SLE patients and that have neutralizing properties. Contrary to previous findings, anti-DNase1L3 IgG associated with presence of anti-dsDNA antibodies and a subpopulation of SLE patients with dual reactivity was identified. While presence of anti-DNase1L3 IgG associated with immunologically more active disease, no

associations were seen with organ disease. Leveraging the available blood transcriptomic data of the SPARE cohort, the study demonstrates increased Interferon signature and myeloid cell activation in most patients positive for anti-DNase1L3 antibodies, and particularly so in those with dual positivity for anti-dsDNA IgG. Importantly, the authors found that a subset of antibodies to DNase1L3 arise from autoreactive VH4-34-expressing IgG switched memory B cells but can also arise from non-autoreactive B cells. These studies are particularly important as they introduce another layer of serologic complexity in SLE that could further subgroup the patients to better understand the pathogenic mechanisms relevant to these subgroups. Several corrections and additional analyses suggested below would strengthen the findings and their interpretation. The techniques used are appropriate and data are largely carefully interpreted with potential for additional analyses. Discussion of the data in the context of prior literature is somewhat lacking.

1. There are discrepancies in numbers presented in the Results section for Figs. 1C-D. The numbers in the figure panels are different as they are in the Supplemental Table 1. All the numbers should be carefully reviewed and corrected. (e.g. the results section indicates 7/113 anti-dsDNA+ patients in the DNase1L3 negative group, fig. 1 shows 14/110 and Suppl. Table 1 shows 58/110). If all these numbers don't indicate #of anti-dsDNA+ patients it should be clarified.
2. Differences in associations or lack thereof with anti-dsDNA IgG in this and the previous study by Hartl et al 2021 are not discussed. Is this driven by the cohort or technical approaches?
3. The SPARE cohort in reference #32 has 95 patients. It is unclear if additional transcriptomic data were generated for the analyses presented in Fig. 3.
4. High IFN and myeloid activation modules are also found in a subpopulation of anti-DNaseIL3 negative patients (Fig. 3E), despite absence of anti-dsDNA IgG. Do these patients have lower disease activity despite myeloid cell activation?
5. Unlike in the present study, Hartl et al showed that anti-DNaseIL3 antibodies were particularly elevated in SLE patients with renal disease. Do differences in organ involvement emerge if patients are sub-grouped depending on co-existence of anti-dsDNA positivity or enrichment scores of different modules? These analyses would help elucidate the pathologic implications of the identified serologies, taking into account multiple parameters. Is the organ injury possibly related to the neutralizing potential of anti-DNase1L3 antibodies?
6. The ability of VH4-34 anti-DNase1L3 monoclonal antibodies to suppress the activity of DNase1L3 was not studied. This could be important given their differences in the affinity for dsDNA as well as differential levels of 9G4+ anti-DNase1L3 antibodies in SLE sera.
7. Cross-reactivity with dsDNA is not unique to anti-DNase1L3 antibodies in lupus patients. Is there published evidence that anti-dsDNA IgG also arise during affinity maturation of autoantibodies targeting other antigens?
8. While it may be beyond the scope of this study, it would be interesting to investigate IgA reactivity to DNase1L3 as anti-dsDNA IgA antibodies may contribute to SLE pathogenesis and 9G4+ IgA+ B cells are present in SLE.

Response to reviewer's comments.

We greatly appreciate the comments given, and have modified the manuscript according to the reviewers' suggestions. We believe these changes have greatly strengthened the paper.

Reviewer #1

Comment 1. I suggest changing the title to specify the target antigen of the autoantibodies characterized in the article. I would consider including the name of anti-DNase1L3, rather than mentioning a "protein antigen".

Response: The title has been modified.

Comment 2. The paragraph "A subgroup of these antibodies is encoded by the inherently autoreactive VH4-34 gene segment, derives from anti-DNase1L3 germline-encoded precursors, and gain cross-reactivity to dsDNA after somatic hypermutations. In a second subgroup, originally defined as nephritogenic anti-dsDNA antibodies, dual reactivity to DNase1L3 and dsDNA is gained during affinity maturation, but their binding efficiencies favor DNase1L3 as the primary antigen" would benefit from rewording to better describe the results.

Response: We are not sure about the problem with the abstract. Maybe the subgroups aren't clear, because they aren't really subgroups. We have rephrased the abstract to define the monoclonals as "groups" instead of "subgroups". Also, the cross-reactivity with cardiolipin and the diverse VH origin of the second group of antibodies were not mentioned because limitations in the word number. The abstract has been modified as follow: "A group of these antibodies is encoded by the inherently autoreactive V_H4-34 gene segment, derives from anti-DNase1L3 germline-encoded precursors, and gains cross-reactivity to dsDNA – and some to cardiolipin – after somatic hypermutation. A second group, originally defined as nephritogenic anti-dsDNA antibodies, is encoded by diverse V_H gene segments. Although affinity maturation results in dual reactivity to DNase1L3 and dsDNA, their binding efficiencies favor DNase1L3 as the primary antigen.". This change increased the abstract from 150 to 158 words. We hope that the Editor will overlook the change in word count.

Comment 3. The heterogeneity of anti-dsDNA antibodies should be better elucidated. Previous studies indeed have investigated the different cross-reactivity of these autoantibodies; please consider providing a description of the possible cross-reactive antigens.

Response: In addition to the references used to support the different cross-reactivity of anti-dsDNA antibodies, some examples of cross-reactive antigens have been added to the introduction, lanes 74-77: "For instance, a subset of anti-dsDNA antibodies that bind the N-methyl-D-aspartate receptor can drive neuronal death and neuropsychiatric lupus¹¹, and cross-reactivity with intrinsic renal antigens, such as α -actinin, has been proposed as a mechanism by which a subset of anti-dsDNA antibodies can mediate nephritis^{15, 19}."

Comment 4. DNase is a limiting factor also for NET accumulation and thus has an important preventive role in the pathogenesis of SLE. I suggest mentioning in the introduction section the role of NETosis in the pathogenesis of the disease with regard to DNase inhibition by auto-antibodies. References should be included.

Response: While DNase 1 has been linked to the degradation of NETs in SLE (Hakim et al. PNAS 2010, 107:9813-9818), to our knowledge, a direct effect of DNase1L3 on NETs degradation or a role

of DNase1L3 in NETs degradation in SLE has not been experimentally addressed. The only publication attempting to link NETs and DNase1L3 exclusively included the analysis of circulating levels of DNase1 and DNase1L3 compared to NETs levels, and DNase activity in serum regardless of the DNase specificity (Bruschi et al. J Rheumatol 2020, 47:377-386). Indeed, all experimental work that we know has been focused on addressing the mechanistic role of DNase1L3 in degradation of DNA from apoptotic cells, but not from NETs. Unrelated to SLE but pertinent to this point, a recent study failed to demonstrate a role of anti-DNase1L3 antibodies in NETs degradation in patients with hidradenitis suppurativa (Oliveira et al. J Invest Dermatol 2023, 143:57-66). Instead, the study found that NET degradation was dependent on DNase1, which, rather than supporting, calls into question the role of DNase1L3 in NET clearance. In this study, the title does not reflect the experimental data and therefore, it needs a careful analysis.

We certainly agree that autoantibody-mediated DNase1 inhibition is a key mechanism in the accumulation of NETs in SLE. However, while this idea could easily be extended to anti-DNase1L3 antibodies, there is no experimental evidence to support it. We are concerned that the manuscript may be misinterpreted by implying an association between DNase1L3 and NETs that does not exist. Therefore, we would prefer to avoid this topic, which has no impact in the manuscript.

Comment 5. Fig 1. I suggest modifying the figure title “Prevalence and clinical associations of antibodies to DNase1L3 in a prospective observational cohort of patients with SLE.” including also the serologic associations.

Response: The figure legend has been modified accordingly: “Prevalence, clinical and serologic associations of antibodies to DNase1L3 in a prospective observational cohort of patients with SLE”

Comment 6. Lines 167-170: the authors define the effect of anti-dsDNA and anti-DNase autoantibodies as “additive” on IFN modules. I suggest reformulating the phrase since the experiments carried out do not explain the direct effect of autoantibodies on IFN modules.

Response: The text has been modified, lines 176-178 in the tracked version: “These findings strongly suggest that the presence of both antibodies has an additive effect on the production of IFN in SLE.”

Comment 7. Fig 2.B Please add the text: orange green red over the clusters in order to better understand figure 2D. Add in the legend what the 3 clusters mean.

Response: To facilitate the interpretation of Figure 2, we decided to leave the colors and term each cluster with a number (Cluster One, Two and Three). Each cluster number has been added to Figure 2B and 2D, and “Cluster Three” has been added to the results section, line 150 in the tracked version. The meaning of the clusters has been added to the legend in Figure 2: “SLE patients clustered in three major groups defined by upregulation of genes related to mRNA processing and translation (Cluster One), upregulation of mRNA processing, translation, and some IFN-stimulated genes (ISGs) (Cluster Two), and upregulation of ISGs and neutrophil activation genes (Cluster Three).”

Comment 8. In addition, I would suggest describing better in the results section and then commenting in the discussion section on the role of increased neutrophil count in anti-DNase1L3 positive versus negative patients. Otherwise, I suggest moving the result to the supplementary section.

Response: Elevated neutrophil count has been associated with higher disease activity in SLE (Han et al. Lupus Sci Med 2020, 7:e000382). In our study, the higher neutrophil count, together with other

clinical and serologic data, provides evidence that anti-DNase1L3 antibodies are related to more active disease in SLE. This information has been added to the result section, lanes 154-155 in the tracked version: "...supporting the association of anti-DNase1L3 antibodies with higher disease activity in SLE ³⁶." The paper by Han et al. was also added to the references (Ref. 36).

Comment 9. What could be the clinical use of characterizing the specificity of anti-dsDNA? Would it be possible to identify nephritogenic antibodies to better treat or follow up patients at risk of nephritis?

Response: Anti-dsDNA antibodies are highly heterogeneous. Having a tool to identify pathogenic autoantibody subsets within the pool of anti-dsDNA antibodies is essential to improve the treatment and follow up of patients with SLE.

Comment 10. Do the authors consider their results relevant also from a diagnostic/therapeutic point of view? Could the blockade or the depletion of anti-DNase be a potential different therapeutic target in Lupus or Lupus Nephritis patients?

Response: The potential diagnostic point was addressed in comment 9 above. About therapeutics, certainly, depletion of anti-DNase1L3 antibodies is a potential therapeutic target in SLE.

Comment 11. I suggest better discussing the cross-reactivity with cardiolipin. Would it have a clinical impact on the patients? Do the patients also have a pro-thrombotic phenotype?

Response: The potential significance of cross-reactivity with cardiolipin has been added to the discussion section, lanes 389-395 in the tracked version: "The finding that some monoclonal antibodies also cross-react with cardiolipin is intriguing, particularly because these antibodies target cardiolipin bound to B2GPI and therefore have the potential to be pathogenic. Although this study did not focus on the analysis of anti-phospholipid syndrome, it is worth noting that anti-DNase1L3 antibody positivity was significantly associated with livedo, lupus anticoagulant, antibodies to cardiolipin and B2GPI, and marginally with arterial thrombosis. Thus, it is possible that triple reactive antibodies – targeting DNase1L3, dsDNA and cardiolipin – may contribute to the pool of anti-phospholipid antibodies in SLE."

Comment 12. Please provide a description of how cardiolipin was tested for anti-cardiolipin reactivity detection. Did the authors use cardiolipin alone or with the protein co-factor (beta-2glycoprotein?)?

Response: Anti-cardiolipin antibodies were detected by ELISA using the QUANTA lite® ACA IgG III assay (Inova Diagnostics, Cat: 708625). This information is found in lanes 534-535 in the tracked version. According to the manufacturer, the microwell plate is coated with both purified cardiolipin and B2GPI. As suggested by the reviewer, it is important to clarify that the antibodies are detecting cardiolipin bound to B2GPI. This information has been added to results, discussion and methods sections, lanes 238, 243, and 534.

Comment 13. The approach is valid and comprehensive of any statistical tests used for the analysis.

Response: Thanks.

Comment 14. Please delete ref 19 as it has not been published yet.

Response: Reference 19 (reference 20 in the revised manuscript) is a preprint deposited in *medRxiv* in June 2021, which describes the discovery of anti-DNase1L3 antibodies by our group.

Unfortunately, at the time that our paper was submitted for peer review in May 2021, another group published a paper describing the same autoantibodies (Hartl, JEM 2021 May 3; 218: e20201138). While our manuscript was timely and contained innovative clinical, transcriptional and mechanistic data distinct from the Hartl paper, it was considered obsolete and no longer original by the same reviewer in two different journals. Nearly two years after our initial description of anti-DNase1L3 antibodies, we chose not to pursue the publication addressing the discovery of the antibodies and instead concentrate on other novelties. The preprint, however, contains essential information about the rationale that led to the discovery of these autoantibodies, as well as experimental evidence of major caveats and misinterpretation in current assays used to quantify DNase1L3 activity in serum and plasma. We consider that this information is important, and that investigators should value it independently, regardless of the opinion of one reviewer. Therefore, we would like to suggest that the Editor decide whether this reference can be cited.

Reviewer #2

Comment 1. This is an outstanding original paper, demonstrating for the first time the existence of autoantibodies in SLE that target both dsDNA and the endonuclease DNase1L3. This subset of cross-reacting autoantibodies exhibit stronger pathogenicity. The data suggest that DNase1L3 may be the primary target of autoantibodies that cross-react with dsDNA. The results, which were methodologically well validated, change the perspective on the origin of anti-dsDNA autoantibodies and their pathogenic role in SLE.

Response: We greatly appreciate the reviewer comments.

Reviewer #3

Comment 1. There are discrepancies in numbers presented in the Results section for Figs. 1C-D. The numbers in the figure panels are different as they are in the Supplemental Table 1. All the numbers should be carefully reviewed and corrected. (e.g. the results section indicates 7/113 anti-dsDNA+ patients in the DNase1L3 negative group, fig. 1 shows 14/110 and Suppl. Table 1 shows 58/110). If all these numbers don't indicate #of anti-dsDNA+ patients it should be clarified.

Response: The error in the number of anti-dsDNA+ patients has been corrected, lanes 132-133 in the tracked version. The correct number of anti-dsDNA positive patients was shown in the figure panels, but was copied incorrectly in the text. The results and conclusions are not affected by this mistake.

The number of anti-dsDNA positive patients varies between the Supplemental Table 1 and Figures 1C-D because the Table shows the frequency and odds ratio of patients ever being positive for anti-dsDNA antibodies, whereas Figure 1C-D shows the frequency of anti-dsDNA antibodies at the time the serum sample was collected.

Comment 2. Differences in associations or lack thereof with anti-dsDNA IgG in this and the previous study by Hartl et al 2021 are not discussed. Is this driven by the cohort or technical approaches?

Response: Hartl et al. used several methods to detect anti-DNase1L3 antibodies, including a bead array assay, which is technically similar to our detection assay. Therefore, the most likely is that we are detecting the same antibodies. The cohorts are different, but unlikely to explain the differences.

Rather, an association between anti-dsDNA and anti-DNase1L3 should be expected to be more significant in the study by Hartl et al. than in our study. Importantly, the study by Hartl et al. neither found an association between anti-DNase1L3 and anti-C1q antibodies, which are highly prevalent in patients with lupus nephritis.

In the methods section, Hartl et al described that their cohort included 120 patients, 72% of whom had renal disease (87/120). Anti-DNase1L3 antibodies were only analyzed in 57/120 patients, of which 70% (40/57) had renal involvement (Figure 2A in the Hartl et al paper). Using a homemade bead-based Ag array, levels of antibodies to dsDNA and C1q were only determined in 33/120 (28%) and 25/120 (21%) patients, respectively (Figure S2D and E in Hartl et al). Moreover, values to define seropositivity to anti-dsDNA and anti-C1q using the bead-based assay were not determined in the manuscript. In our cohort, 48% of patients had renal disease (75/158, 48%), which is closer to the reported prevalence of nephritis in SLE (~40%). In addition, we did not preselect patients for our analyzes, allowing to have a heterogenous and more real sample of patients with SLE.

Considering that antibodies to dsDNA and C1q are present in up to 80% and 100% of patients with renal disease, respectively (Stojan and Petri. *Lupus* 2016; 25:873-7), and that 72% of the patients had renal involvement, it is surprising that Hartl et al. found no associations of these antibodies with anti-DNase1L3 antibodies, unless a significant number of the patients with renal disease were seronegative for antibodies to dsDNA and C1q. Our ability to address this hypothesis is limited by the absence of information in the methods used to select the patient subgroups for autoantibody analysis. Alternatively, Hartl et al. only examined the correlation between autoantibody levels, which was not significant (Figure S2D and E in Hartl et al), rather than the association with antibody positivity, as we did in our study. For the correlation analysis, Hartl et al. only included small subsets of patients in which the autoantibodies were detected using the homemade bead-based antigen array [i.e. anti-dsDNA in 33/120 (28%) and anti-C1q in 25/120 (21%) patients]. Aside from the small sample size, finding a significant correlation between antibody levels is unlikely because cross-reactive anti-DNase1L3/dsDNA antibodies only correspond to a subset of antibodies detected by the anti-DNase1L3 and anti-dsDNA assays. The most likely is that the study by Hartl et al. also has the same association with anti-dsDNA positivity, but they did not look for it. Unfortunately, besides the presence of renal involvement, the study by Hartl et al provides no information about anti-dsDNA positivity (or any other clinical or serological parameter) that may help to confirm this notion.

The study by Hartl et al., however, also included something termed “DNase1L3-sensitive microparticle (MP) antigen assay”, which appears to define the capacity of autoantibodies (such as anti-dsDNA) to bind to MP antigens sensitive to DNase1L3 degradation (Figure 6 in Hartl et al.). Although the assay was not directly compared with levels or positivity of anti-DNase1L3 antibodies, it is described as an indirect assay to determine the functional consequences of DNase1L3 inhibition by anti-DNase1L3 antibodies in SLE. By combining SLE patients with (n = 61) or without (n = 59) renal disease, Hartl et al. found that reactivity to DNase1L3-sensitive antigens on MPs was significantly associated with positivity to anti-dsDNA antibodies (likely determined by ELISA). This association was independent of renal disease (Figure 6H, right graph, in Hartl et al.). We are not sure how to interpret these data in the context of anti-DNase1L3 antibodies, but if Hartl et al. are correct regarding the functional implication of the assay, these findings indirectly confirm that anti-DNase1L3 antibodies are associated with antibodies to dsDNA in SLE.

Please, also see the response to comment 5 below to complement this response.

Comment 3. The SPARE cohort in reference #32 has 95 patients. It is unclear if additional transcriptomic data were generated for the analyses presented in Fig. 3.

Response: The SPARE cohort comprises of 306 SLE patients who were evaluated at baseline and throughout the course of two years in accordance with their requirement for clinical care. Every patient has at least one visit with determination of gene expression by microarray. Reference #32 (reference 33 in the revised manuscript) corresponds to a sub-analysis of 95 SLE patients of the cohort. In our study, 158 SLE patients with accessible serum and microarray data were randomly selected from SPARE. Other than the ones accessible for the SPARE cohort, no further gene expression experiments were carried out for this study.

Comment 4. High IFN and myeloid activation modules are also found in a subpopulation of anti-DNAse1L3 negative patients (Fig. 3E), despite absence of anti-dsDNA IgG. Do these patients have lower disease activity despite myeloid cell activation?

Response: The reviewer is correct. This subgroup has lower disease activity, as shown in the bar graphs.

Comment 5. Unlike in the present study, Hartl et al showed that anti-DNAse1L3 antibodies were particularly elevated in SLE patients with renal disease. Do differences in organ involvement emerge if patients are sub-grouped depending on co-existence of anti-dsDNA positivity or enrichment scores of different modules? These analyses would help elucidate the pathologic implications of the identified serologies, taking into account multiple parameters. Is the organ injury possibly related to the neutralizing potential of anti-DNAse1L3 antibodies?

Response: In addition to anti-DNAse1L3 antibodies, the only clinical and serological features studied by Hartl et al. were renal involvement and levels (not positivity) of antibodies to dsDNA and C1q. Importantly, these parameters were only determined in patients' subsets within their original cohort: anti-DNAse1L3 antibodies in 57/120 (48%), anti-dsDNA antibodies in 33/120 (28%) and anti-C1q antibodies in 25/120 (21%) (Figures 2A, S2D and S2E, respectively, in Hartl et al). In addition, Hartl et al. used a cohort that was highly enriched in patients with renal disease (72%). Hartl et al. only found an association of anti-DNAse1L3 antibodies with renal disease because it was the only clinical feature that was analyzed in their study. Considering that anti-DNAse1L3 antibodies were only determined in 57 SLE patients, which included 70% with renal disease (Figure 2A in Hartl et al.), the number of patients is limited to search for other significant associations. In our study, 60% (29/48) of anti-DNAse1L3 positive patients had renal involvement compared to 41.8% (46/110) patients negative for these antibodies ($P = 0.038$) (Supplemental Table 1). Thus, regarding the association of these autoantibodies with renal disease, our results are not different to study by Hartl et al. Other discrepancies are explained by differences in the clinical heterogeneity of the cohorts (renal disease in Hartl et al. 70% vs. Hopkins 48%), the number of patients who were tested for anti-DNAse1L3 antibodies (Hartl et al. $n = 57$ vs. Hopkins $n = 158$), and the detail description and analysis of clinical and serological associations with anti-DNAse1L3 antibodies. Additional discussion on this point is included in the response to comment 2 above.

The responses to comments 2 and 5 were summarized in the discussion section, lines 315-336 in the tracked version.

Comment 6. The ability of VH4-34 anti-DNAse1L3 monoclonal antibodies to suppress the activity of DNAse1L3 was not studied. This could be important given their differences in the affinity for dsDNA as well as differential levels of 9G4+ anti-DNAse1L3 antibodies in SLE sera.

Response: The effect of VH4-34 anti-DNase1L3 monoclonal antibodies on DNase1L3 activity was shown in Figure 4D.

Comment 7. Cross-reactivity with dsDNA is not unique to anti-DNase1L3 antibodies in lupus patients. Is there published evidence that anti-dsDNA IgG also arise during affinity maturation of autoantibodies targeting other antigens?

Response: The short answer is not to our knowledge. The discovery that anti-DNase1L3 antibodies cross-react with dsDNA, which led to the analysis of germline reverted antibodies, was coincidental. The antigenic analysis of monoclonal autoantibodies reverted to germline is usually centered on the reactivity against the antigen used to identify the antibodies (i.e. the “primary” antigen). This type of analysis has led to the major conclusion that autoantibodies develop from nonreactive precursors by SHM (Mietzner et al. PNAS 2008,105:9727-32). However, whether germline reverted autoantibodies have the potential to bind any other antigen, besides their original target, has not been addressed.

Comment 8. While it may be beyond the scope of this study, it would be interesting to investigate IgA reactivity to DNase1L3 as anti-dsDNA IgA antibodies may contribute to SLE pathogenesis and 9G4+ IgA+ B cells are present in SLE.

Response: This is an excellent point. We will follow this suggestion in further studies.

REVIEWERS' COMMENTS

Reviewer #1 (Remarks to the Author):

The authors very carefully addressed the comments made and reviewed some confusing or unclear elements of the manuscript.

Reviewer #3 (Remarks to the Author):

The authors addressed all the comments. This is an exciting and valuable study for our understanding of lupus pathogenesis.